# Steady State Analysis of Episodic Reinforcement Learning

**Huang Bojun**
Rakuten Institute of Technology, Tokyo, Japan
`bojhuang@gmail.com`

## Abstract

This paper proves that the episodic learning environment of every finite-horizon decision task has a unique *steady state* under any behavior policy, and that the marginal distribution of the agent's input indeed converges to the steady-state distribution in essentially all episodic learning processes. This observation supports an interestingly reversed mindset against conventional wisdom: While the existence of unique steady states was often presumed in continual learning but considered less relevant in episodic learning, it turns out their existence is *guaranteed* for the latter. Based on this insight, the paper unifies episodic and continual RL around several important concepts that have been separately treated in these two RL formalisms. Practically, the existence of unique and approachable steady state enables a general way to collect data in episodic RL tasks, which the paper applies to policy gradient algorithms as a demonstration, based on a new *steady-state policy gradient theorem*. Finally, the paper also proposes and experimentally validates a perturbation method that facilitates rapid steady-state convergence in real-world RL tasks.

## 1 Introduction

Given a decision task, reinforcement learning (RL) agents iteratively optimize decision policy based on empirical data collected from continuously interacting with a learning environment of the task. To have successful learning, the empirical data used for policy update should represent some desired distribution. Every RL algorithm then faces two basic questions: *what* data distribution shall be considered desired for the sake of learning, and *how* should the agent interact with the environment to actually obtain the data as desired? Depending on the type of the given task, though, existing RL literature have treated this data collection problem in two different ways.

In *continual reinforcement learning*, the agent immerses itself in a single everlasting sequential-decision episode that is conceptually of infinite length (or of life-long length). In this case, a *stationary (or steady-state) distribution* is a fixed-point distribution over the agent's input space under the transition dynamic induced by a decision policy. The concept of steady state has been pivotal in continual RL literature. It is typically *presumed* that a unique stationary distribution exists when rolling out any policy in the continual RL model being studied, and that the empirical data distribution of the rollout indeed converges over time to this stationary distribution due to (again assumed) ergodicity of the system dynamic [28, 3, 23, 35]. Continual RL algorithms can be derived and analyzed by examining system properties under the steady state [10, 28, 27, 34], and many of the resulted algorithms require the training data to follow the stationary distribution of some behavior policy [3, 23] (or a mixture of such distributions [12]), which can then be efficiently collected from a few (or even a single) time steps once the rollout of the behavior policy converges to its steady state.

The situation has been quite different for *episodic reinforcement learning*, in which the agent makes a finite number of decisions before an episode of the task terminates. Episodic RL tasks account for the vast majority of experimental RL benchmarks and of empirical RL applications at the moment [2, 14].

Due to the finiteness of decision horizon, for episodic tasks steady state was widely considered non-existent [27] or having a degenerate form [23], in either case irrelevant. The lack of meaningful steady state in such tasks has in turn led to more modeling disparities in the episodic RL formalism. Indeed, as Sutton and Barto [23] (chap. 9, p. 200) wrote: "*[t]he two cases, continuing and episodic, behave similarly, but with [function] approximation they must be treated separately in formal analyses, as we will see repeatedly in this part of the book*". In particular, the desired data distributions for episodic RL algorithms are usually characterized by alternative concepts, such as the expected *episode-wise visiting frequencies* [23], or a $\gamma$-discounted variant of it [19, 15].

Intriguingly, despite the rather different theoretical framework behind, many episodic RL algorithms [13, 23] update policies using data from a small time slice (such as from one time step [23] or from several consecutive steps [13, 17]) of the episodic learning process, in almost the same way as how continual RL algorithms [31, 13] collect data from ergodic models. This "online-style" data collection [4], while being popular in modern RL algorithms[13, 17, 12, 9, 7] thanks to its simplicity and sample efficiency, is often not well justified in episodic tasks by the prescribed data distribution which requires collecting a whole episode trajectory as *one* sample point of the distribution [4].

In this paper, we propose a new perspective to treat the data collection problem for episodic RL tasks. We argue that in spite of the finite decision horizon, the RL agent has the chance to *repeat* the same decision process in the *learning environment* of an episodic RL task. The resulted learning process is an infinite loop of homogeneous and finite episodes. By formalizing such an episodic learning environment into a family of infinite-horizon MDPs with special structures (Section 3), we mathematically proved that a unique stationary distribution exists in every such episodic learning process, regardless of the agent policy, and that the marginal distribution of the rollout data indeed converges to this stationary distribution in "proper models" of the episodic RL task (Section 4).

Conceptually, our theoretical observation supports a reversed mindset against conventional wisdom: while unique steady state is (only) presumed to exist in continual tasks, it is now *guaranteed* to exist in all episodic tasks. Moreover, through analyzing the Bellman equation under the steady state, we obtained interesting and rigorous connections between the separately defined semantics of important concepts (such as *performance measure* and *on-policy distribution*) across episodic and continual RL. These results help with unifying the two currently disparate RL formalisms (Section 5).

Algorithmically, the unified framework enables us to collect data efficiently in episodic tasks in a similar way as in the continual setting: Write the quantity to be computed as an expectation over the stationary distribution of some policy $\beta$, rollout $\beta$ and wait for it to converge, after which we obtain an unbiased sample point in every single time step. As a demonstration of this general approach, we derived a new *steady-state policy gradient theorem*, which writes policy gradient into such an expected value. The new policy gradient theorem not only better justifies the popular few-step-sampling practice used in modern policy gradient algorithms [13, 23], but also makes explicit some less explored bias in those algorithms (Section 6.1). Finally, to facilitate *rapid* steady-state convergence, we proposed a *perturbation trick*, and experimentally showed that it is both necessary and effective for data collection in episodic tasks of standard RL benchmark (Section 6.2).

## 2 Preliminaries

**Markov chain**. In this paper, a *markov chain* is a homogeneous discrete-time stochastic process with countable (finite or infinite) state space. The state-transition probabilities are written into a *transition matrix* $M$, where $M(s, s')$ is the entry of row $s$ and column $s'$ which specifies $\mathbb{P}[s_{t+1} = s'|s_t = s]$. A *rollout* of $M$ generates a trajectory $\zeta = (s_0, s_1, s_2, \dots)$ of infinite length. Let row vector $\rho^{(t)}$ denote the marginal distribution of the state at time $t$, so $s_t \sim \rho^{(t)} = \rho^{(t-1)}M$. The *limiting distribution* of $s_t$, if exists, is the marginal distribution at infinite time, i.e. $\lim_{t \to \infty} \rho^{(t)}$, and the *stationary (or steady-state) distribution* is defined as the fixed-point distribution with $\rho = \rho M$.

The existence and uniqueness of stationary and limiting distributions are characterized by the following well-known concepts and conditions: Given a markov chain with transition matrix $M$, a state $s$ is *reachable* from a state $\bar{s}$ if $\mathbb{P}[s_t = s|s_0 = \bar{s}] > 0$ for some $t \geq 1$, and the markov chain $M$ is said *irreducible* if every state is reachable from any state in $M$. The *mean recurrence time* of a state $s$ is the expected number of steps for $s$ to reach itself, denoted as $\mathbb{E}[T_s] \doteq \sum_{t=1}^{\infty} t \cdot \mathbb{P}[s_t = s$ and $s \notin \{s_{1:t-1}\}|s_0 = s]$, and state $s$ is said *positive recurrent* if $\mathbb{E}[T_s] < \infty$. The markov chain $M$ is positive recurrent if every state in $M$ is positive recurrent. Finally, a state $s$ is *aperiodic* if $s$ can reach itself in two trajectories with co-prime lengths, i.e. if $\gcd\{t > 0 : \mathbb{P}[s_t = s|s_0 = s] > 0\} = 1$.

**Proposition 1.** *An irreducible markov chain $M$ has a unique stationary distribution $\rho_M = 1/\mathbb{E}[T_s]$ if and only if $M$ is positive recurrent. ([18], Theorem 54)*

**Proposition 2.** *An irreducible and positive-recurrent markov chain $M$ has a limiting distribution $\lim_{t \to \infty} \rho^{(t)} = \rho_M$ if and only if there exists one aperiodic state in $M$. ([18], Theorem 59)*

A markov chain satisfying the condition in Proposition 2 is called an *ergodic* markov chain.

**Markov Decision Process (MDP).** An MDP $\mathcal{M} = (\mathcal{S}, \mathcal{A}, R, P, \rho_0)$ is a sequential decision making model where $\mathcal{S}$ is a countable state space, $\mathcal{A}$ a countable action space, $R(s) \in [r_{min}, r_{max}]$ is a reward assigned to each state $s \in \mathcal{S}$, $P(s'|s,a) \doteq \mathbb{P}[s_{t+1} = s'|s_t = s, a_t = a]$ specifies action-conditioned transition probabilities between states, and $\rho_0$ is the initial distribution with $s_0 \sim \rho_0$.

A policy function $\pi : \mathcal{S} \times \mathcal{A} \to [0,1]$ prescribes the probability to take action under each state. With $\mathbb{P}[a_t = a|s_t = s] = \pi(s,a)$, every policy $\pi$ induces a markov chain with transition matrix $M_\pi(s,s') = \sum_a \pi(s,a) P(s'|s,a)$. Accordingly, row vectors $\rho_\pi^{(t)}$ and $\rho_\pi$ denote the marginal and stationary distributions of $s_t$ under policy $\pi$, respectively, so $\rho_\pi^{(t+1)} = \rho_\pi^{(t)} M_\pi$, and $\rho_\pi = \rho_\pi M_\pi$.

An MDP is ergodic if for every policy $\pi$ the induced markov chain $M_\pi$ is ergodic, in which case the *steady-state performance* [28] of a policy $\pi$ is $J_{avg}(\pi) \doteq \lim_{T \to \infty} \frac{1}{T} \sum_{t=1}^{T} R(s_t) = \mathbb{E}_{s \sim \rho_\pi}[R(s)]$. When a set of *terminal states* $\mathcal{S}_\perp \subset \mathcal{S}$ is identified, a rollout trajectory $\zeta$ is said *terminated* when it reaches a terminal state. We use $T(\zeta) \doteq \inf\{t \geq 1 : s_t \in \mathcal{S}_\perp\}$ to denote the *termination time* of $\zeta$, and the *episode-wise performance* of a policy $\pi$ is defined as $J_{epi}(\pi) \doteq \mathbb{E}_{\zeta \sim M_\pi}[\sum_{t=1}^{T(\zeta)} R(s_t)]$.

A value function $Q : \mathcal{S} \times \mathcal{A} \to \mathbb{R}$ prescribes the "value" of taking an action under a state. In this paper, we consider the following *product-form family* of value functions [25, 30, 3, 21, 26, 20, 32] :

$$Q_\pi(s,a) \doteq \mathbb{E}_{s_1 \sim P(s,a),\{s_{\geq 2}\} \sim M_\pi} [\sum_{t=1}^{\infty} R(s_t) \cdot \prod_{\tau=1}^{t-1} \gamma(s_\tau)] \tag{1}$$

$$= \mathbb{E}_{s' \sim P(s,a), a' \sim \pi(s')} [R(s') + \gamma(s') \cdot Q_\pi(s',a')] \tag{2}$$

in which $P(s,a)$ and $\pi(s')$ are short-hands for the conditional distributions $P(\cdot|s,a)$ and $\pi(\cdot|s')$, and $\gamma : \mathcal{S} \to [0,1]$ is called the *discounting function*. Besides entailing the *Bellman equation* (2), the definition of $Q_\pi$, i.e. (1), also induces the *state-value function* $V_\pi(s) \doteq \mathbb{E}_{a \sim \pi(s)}[Q_\pi(s,a)]$.

With $\gamma(s) = \gamma_c$ for constant $\gamma_c \in [0,1)$, the product-form value functions $Q_\pi$ and $V_\pi$ subsume the classic value functions that have been underlying much of the existing RL literature, and we will use $Q_\pi^{\gamma_c}$ and $V_\pi^{\gamma_c}$ to refer to this particular form of value functions with constant discounting function. On the other hand, with $\gamma(s) = \mathbb{1}(s \notin \mathcal{S}_\perp)$, the episode-wise performance $J_{epi}$ can be recast as the product-form value of the initial state [25]: $J_{epi}(\pi) = V_\pi(s_0)$ if $\gamma(s) = 1$ for $s \notin \mathcal{S}_\perp$ and $\gamma(s) = 0$ for $s \in \mathcal{S}_\perp$. We will call the function $\gamma(s) = \mathbb{1}(s \notin \mathcal{S}_\perp)$, the *episodic discounting function*. See Appendix A for more discussions on the formulation choices behind the MDP model.

**Reinforcement Learning (RL).** In continual RL, we are given a decision task modeled by an ergodic MDP $\mathcal{M}_D$, and the goal is to find good policy with respect to the steady-state performance $J_{avg}$ through a single rollout of $\mathcal{M}_D$. In episodic RL, the decision task $\mathcal{M}_D$ is a finite-horizon MDP with specified terminal states, and the goal is to optimize the episode-wise performance $J_{epi}$ through repeated rollouts of $\mathcal{M}_D$. In this case, a special resetting procedure will intervene the learning process to start a new decision episode upon the end of the last [2].

A common and basic idea, for both continual and episodic RL algorithms, is to use the rollout data to optimize a parameterized policy function $\pi(s,a;\theta)$ with respect to a surrogate objective $\tilde{J}(\theta)$ via the stochastic gradient method. At a update time $t$, the gradient $\nabla_\theta \tilde{J}$ is approximately computed as a sample mean of some computable function $F$ over a "mini-batch" $\mathcal{D}_t$, with $\nabla_\theta \tilde{J}(\theta_t) \approx \frac{1}{|\mathcal{D}_t|} \sum_{(s,a) \in \mathcal{D}_t} F(s,a,\theta_t)$, where $\mathcal{D}_t$ is a selected subset of all the rollout data up to time $t$.

For example, *policy gradient algorithms*, as a family of widely used RL algorithms, work on policy functions $\pi(s,a;\theta)$ that are directly differentiable, and typically choose $\tilde{J}_{pg}(\theta) \doteq \mathbb{E}_{s_0 \sim \rho_0}[V_{\pi(\theta)}^{\gamma_c}(s_0)]$ [1] in episodic tasks as the surrogate objective, whose gradient can be estimated by $F_{pg}(s,a,\theta) \doteq \widehat{Q_\theta^{\gamma_c}}(s,a) \nabla \log \pi(s,a;\theta)$, where $\widehat{Q_\theta^{\gamma_c}}(s,a)$ is some practical estimation of $Q_\theta^{\gamma_c}(s,a)$ for the specific

$(s, a) \in \mathcal{D}_t$. Substituting $\tilde{J}_{pg}$ and $F_{pg}$ to the general RL algorithmic framework above, yields

$$\nabla J_{epi}(\theta) \approx \nabla \tilde{J}_{pg}(\theta) = \sum_{\tau=0}^{\infty} (\gamma_c)^{\tau} \underset{s_{\tau} \sim \rho_{\theta}^{(\tau)}, \, a_{\tau} \sim \pi(s_{\tau};\theta)}{\mathbb{E}} \left[ Q_{\theta}^{\gamma_c}(s_{\tau}, a_{\tau}) \cdot \nabla \log \pi(s_{\tau}, a_{\tau}; \theta) \right] \quad (3)$$

$$\approx \frac{1}{|\mathcal{D}_t|} \sum_{(s,a) \in \mathcal{D}_t} \widehat{Q_{\theta}^{\gamma_c}}(s, a) \cdot \nabla \log \pi(s, a; \theta) \quad (4)$$

where the equality part in (3) is known as the classic *policy gradient theorem* [24].

Policy gradient algorithms illustrate a general disparity observed in episodic RL between the "desired" data distribution and the data actually collected. Specifically, according to (3), the policy gradient should be estimated by rolling out an episode using $\pi(\theta)$ then summing over $(\gamma_c)^{\tau} \cdot F_{pg}$ across all steps $\tau$ in the episode, which collectively provide one sample point to the right-hand side of (3). Many policy gradient algorithms indeed work this way [4, 15]. However, some modern policy gradient algorithms, such as A3C [13] or the variant of REINFORCE as described in [23], compute the policy gradient based on data only from a small time window, much smaller than the average episode length. Such an "online" approach [4] has witnessed remarkable empirical success in practice [13].

A popular explanation for the disparity above is to re-arrange the sum $\sum_{\tau=0}^{\infty} (\gamma_c)^{\tau} \sum_{s_{\tau}} \rho_{\theta}^{(\tau)}(s_{\tau})$ in (3) into $\sum_s \left( \sum_{\tau=0}^{\infty} (\gamma_c)^{\tau} \rho_{\theta}^{(\tau)}(s) \right)$, which is a weighted sum over states, thus can be interpreted as an expectation over a state distribution that we will call it the *episode-wise visitation distribution* $\mu_{\pi}^{\gamma_c}(s) \doteq \sum_{\tau=0}^{\infty} (\gamma_c)^{\tau} \rho_{\pi}^{(\tau)}(s)/Z$ [19, 15, 23], where $Z$ is a normalizing term. However, the visitation-distribution interpretation cannot explain why a *single* rollout data point $s_t$ can serve as a faithful sample of $\mu_{\pi}^{\gamma_c}$. For $\gamma_c < 1$, we know that $s_t$ is just biased to $\mu_{\pi}^{\gamma_c}$ [27]; for $\gamma_c = 1$, we have $\mu_{\pi}^1 \propto \mathbb{E}_{\pi}[\sum_t \mathbb{1}(s_t = s)]$, which justifies the practice of taking long-run average across multiple episodes [4, 15], but still cannot explain why a single step can represent this long-run average [13, 23].

We remark that the aforementioned RL algorithm framework, as well as the disparity between desired data and actual data, apply not only to policy gradient algorithms, but also to other *policy-based* algorithms like PPO[17], and even to *value-based* and *off-policy* algorithms like Q-Learning[31] as well. These algorithms differ "only" in how the policy function $\pi$ is parameterized and in how $\tilde{J}$, $F$, and $\mathcal{D}_t$, are specifically constructed. See Appendix B for more comprehensive discussions.

## 3 Modeling reinforcement learning process with MDP

During the rollout of a policy $\pi$, the state $s_t$ follows the marginal distribution $\rho_{\pi}^{(t)}$ that is dynamically evolving as the time $t$ changes. To reliably use such dynamic data to represent whatever *fixed* distribution as desired, we must understand *how* the marginal state distribution $\rho_{\pi}^{(t)}$ will evolve over time in reinforcement learning tasks. The first step, however, is to have a formalism to the actual environment that the rollout data $s_t$ come from and the marginal distributions $\rho_{\pi}^{(t)}$ reside in.

For episodic tasks, it is important to distinguish the *learning environment* of a task, denoted by $\mathcal{M}_L$, from the decision environment of the task, denoted by $\mathcal{M}_D$ – the latter terminates in finite steps, while the former is an infinite loop of such episodes. This difference seems to be recognized by the community [23], yet the tradition mostly focused on formulating the decision environment $\mathcal{M}_D$ as MDP. The multi-episode learning environment $\mathcal{M}_L$, as *the* physical source where learning data really come from for all episodic RL tasks, has received relatively little treatment in the literature.

We adopt an approach similar (in spirit [2]) with [32] to formulate reinforcement learning environments. In general, the *learning process* of a finite-horizon MDP $\mathcal{M}_D$ is an infinite-horizon MDP $\mathcal{M}_L$. We require that a learning process always starts from a terminal state $s_0 \in \mathcal{S}_{\perp}$, followed by a finite steps of rollout until reaching another terminal state $s_{T_0}$. At $T_0$, the learning process gets implicitly reset into the second episode no matter what action $a_{T_0}$ is taken under state $s_{T_0}$, from there the rollout will reach yet another terminal state $s_{T_0+T_1}$ after another $T_1$ steps. In contrast to the case in $\mathcal{M}_D$, a terminal state of the learning MDP $\mathcal{M}_L$ does not terminate the rollout but serves as the initial state of the next episode, thus has identical transition probabilities with the initial state. Also note that in $\mathcal{M}_L$ a terminal state $s \in \mathcal{S}_{\perp}$ is just a normal state that the agent will spend one real step on it in physical time, from which the agent receives a real reward $R(s)$ [2]; different terminal states may yield different rewards (e.g. win/loss). There is no state outside $\mathcal{S}$, and in particular $\mathcal{S}_{\perp} \subseteq \mathcal{S}$.

As the learning process of episodic task is just a standard MDP, all formal concepts about MDP introduced in Section 2 also apply to episodic learning process. We will use the product-form value function (1) to analyze the learning process, which with $\gamma(s) = \mathbb{1}(s \notin \mathcal{S}_\perp)$ still recovers the episode-wise performance; specifically, $J_{epi}(\pi) = V_\pi(s_\perp)$, where $s_\perp$ can be any terminal state.

However, the *interpretations* and roles of $\rho_\pi^{(t)}$, $\rho_\pi$ and $J_{avg}$ have completely changed when we shifted from $\mathcal{M}_D$ to $\mathcal{M}_L$ (which is our main purpose to shift to $\mathcal{M}_L$ in the first place). To prepare theoretical analysis, we need to formally define *episodic learning process* using two natural conditions.

**Definition 1.** *A MDP $\mathcal{M}_L = (\mathcal{S}, \mathcal{A}, R, P, \rho_0)$ is an **episodic learning process** if*

1. *(Homogeneity condition): All initial states have identical and action-agnostic transition probabilities; formally, $\forall s, s' \in \mathcal{S}, a, a' \in \mathcal{A}$, $P(s, a) = P(s', a')$ if $\rho_0(s) \cdot \rho_0(s') > 0$.*

   *Let $\mathcal{S}_\perp \subseteq \mathcal{S}$ be the set of all states sharing the same transition probabilities with the initial states; that is, $\mathcal{S}_\perp$ is the maximal subset of $\mathcal{S}$ such that $\{s : \rho_0(s) > 0\} \subseteq \mathcal{S}_\perp$ and that $\forall s, s' \in \mathcal{S}_\perp, a, a' \in \mathcal{A}, P(s, a) = P(s', a')$. A state $s \in \mathcal{S}_\perp$ is called a **terminal state**. A finite trajectory segment $\xi = (s_t, s_{t+1}, \ldots, s_{t+T})$ is called an **episode** iff $s_t, s_{t+T} \in \mathcal{S}_\perp$ and $s_\tau \notin \mathcal{S}_\perp$ for $t < \tau < t + T$, in which case the **episode length** is the number of transitions, denoted $|\xi| \doteq T$.*

2. *(Finiteness condition): All policies have finite average episode length; formally, for any policy $\pi$ let $\Xi_\pi = \{\xi : \xi \text{ is episode}, \mathbb{P}_\pi(\xi) > 0\}$ be the set of admissible episodes under $\pi$, then $\sum_{\xi \in \Xi_\pi} \mathbb{P}_\pi(\xi) = 1$ (so that $\mathbb{P}_\pi$ is a probability measure over $\Xi_\pi$ [3]) and $\sum_{\xi \in \Xi_\pi} \mathbb{P}_\pi(\xi) \cdot |\xi| < +\infty$.*

As a familiar special case, an RL environment with time-out bound has a maximum episode length, thus always satisfies the finiteness condition (and having proper time-out bound *is* important and standard practice [14]). Another example of episodic learning process is the MDPs that contain self-looping transitions but the self-looping probability is less than 1; in that case there is no maximum episode length, yet the finiteness condition still holds. On the other hand, the learning process of a pole-balancing task without time-out bound is *not* an episodic learning process as defined by Definition 1, as the optimal policy may be able to keep the pole uphold forever, and for that policy the probabilities of all finite episodes do not sum up to 1.

Importantly, while the episodic learning process as defined by Definition 1 indeed encompasses the typical learning environments of all finite-horizon decision tasks, the former is *not* defined as the latter. Instead, any RL process with the two *intrinsic* properties prescribed by Definition 1 is an episodic learning process. In the next section we will see that these two conditions are enough to guarantee nice theoretical properties that were previously typically only assumed in continual tasks.

## 4 Episodic learning process is ergodic

We first prove that episodic learning process admits unique and meaningful stationary distributions.

**Lemma 1.1.** *(Episodic learning process is irreducible.) In any episodic learning process $\mathcal{M}$, for every policy $\pi$, let $\mathcal{S}_\pi$ be the set of states that are reachable under $\pi$, the induced markov chain $M_\pi$ admits a rollout that transits from any $s \in S_\pi$ to any $s' \in S_\pi$ in a finite number of steps.*

*Proof idea:* Due to homogeneity, a reachable state must be reachable in one episode from any terminal state, thus any $s$ can reach any $s'$ in two episodes (through an terminal state). See C.1 for details. □

**Lemma 1.2.** *(Episodic learning process is positive recurrent.) In any episodic learning process $\mathcal{M}$, for every policy $\pi$, let $s \in \mathcal{S}_\pi$ be any reachable state in the induced markov chain $M_\pi$, then $\mathbb{E}_\pi[T_s] < +\infty$ (where $\mathbb{E}_\pi[T_s]$ is the mean recurrence time of $s$, as defined in Section 2).*

*Proof idea:* If the first recurrence of $s$ occurs in the $n$-th episode, then the conditional expectation of $T_s$ in that case is bounded by $n \cdot \mathbb{E}_{\xi \sim \pi} |\xi|$, which is a finite number due to the finiteness condition. More calculations show that the mean of $T_s$ after averaging over $n$ is still finite. See C.2 for details. □

**Theorem 1.** *In any episodic learning process $\mathcal{M}$, every policy $\pi$ has a unique stationary distribution $\rho_\pi = \rho_\pi M_\pi$.*

*Proof.* Lemma 1.1 and Lemma 1.2 show that the Markov chain $M_\pi$ is both irreducible and positive recurrent over $S_\pi$, which gives the unique stationary distribution over $S_\pi$ by Proposition 1. Padding zeros for unreachable states completes the row vector $\rho_\pi$ as desired. $\qquad\square$

From Proposition 2 we also know that the exact probability $\rho_\pi(s)$, for each state $s$, equals the reciprocal of its mean recurrence time $1/\mathbb{E}_\pi[T_s]$. Theorem 1 tells us that *if* an episodic learning process $\mathcal{M}$ converges, it can only converges to a single steady state. But the theorem does not assert that $\mathcal{M}$ will converge at all. In general, the marginal distribution $\rho_\pi^{(t)}$ could be *phased* over time and never converge. Nevertheless, the following lemma shows that minimal diversity/randomness among the lengths of admissible episodes is enough to preserve ergodicity in episodic learning process.

**Lemma 2.1.** *In any episodic learning process $\mathcal{M}$, the markov chain $M_\pi$ induced by any policy $\pi$ converges to its stationary distribution $\rho_\pi$ if $M_\pi$ admits two episodes with co-prime lengths; that is, $\lim_{t\to\infty} \rho_\pi^{(t)} = \rho_\pi$ if there exist different $\xi, \xi' \in \Xi_\pi$ such that $\gcd(|\xi|, |\xi'|) = 1$.*

*Proof idea:* Lemma 1.1 has shown that $M_\pi$ is irreducible over $\mathcal{S}_\pi$, so we only need to identify one *aperiodic* state $s \in \mathcal{S}_\pi$, then Proposition 2 gives what we need. Intuitively, an episode is a recurrence from the terminal subset $\mathcal{S}_\perp$ to the subset itself, so having co-prime episode lengths entails that, if we aggregate all terminal states into one, the aggregated "terminal state" is an aperiodic state. The same idea can be extended to the multi-terminal-state case. See Appendix C.3 for the formal proof. $\qquad\square$

The co-primality condition in Lemma 2.1 might look restrictive at first glance, but there is a simple and general way to slightly modify *any* episodic learning model $\mathcal{M}$ to turn it into an equivalent model $\mathcal{M}^+$ such that $\mathcal{M}^+$ does satisfy the co-primality condition. The idea is to inject a minimum level of randomness to every episode by introducing an single auxiliary state $s_{null}$ so that an episode in the modified model may start either directly from a terminal state to a normal state as usual, or go through a one-step detour to $s_{null}$ (with some probability $\epsilon$) before the "actual" episode begins.

**Definition 2.** *A $\epsilon$-perturbed model of an episodic learning process $\mathcal{M}$ is a MDP $\mathcal{M}^+$ with $\mathcal{M}^+ = (\mathcal{S}^+, \mathcal{A}, R, P^+, \rho_0)$, where $\mathcal{S}^+ = \mathcal{S} \cup \{s_{null}\}$, and with constant $0 < \epsilon < 1$,*

$$P^+(s'|s,a) = \quad
\begin{array}{c|cc}
\diagbox{s \in}{s' \in} & \mathcal{S} & \{s_{null}\} \\
\hline
\mathcal{S}_\perp & (1-\epsilon)P(s'|s,a) & \epsilon \\
\mathcal{S} \setminus \mathcal{S}_\perp & P(s'|s,a) & 0 \\
\{s_{null}\} & P(s'|s_0,a) & 0
\end{array} \quad .$$

*We also prescribe $R(s_{null}) = 0$, $\rho_0(s_{null}) = 0$, $\gamma(s_{null}) = 1$ for the auxiliary state $s_{null}$ in $\mathcal{M}^+$.*

Note that the null state $s_{null}$ can only be reached at the beginning of an episode in a one-shot manner. Also, by the definition of terminal state (in Definition 1), $s_{null}$ is not a terminal state of $\mathcal{M}^+$.

**Theorem 2.** *In the $\epsilon$-perturbed model $\mathcal{M}^+$ of any episodic learning process, every policy $\pi$ has a limiting distribution $\lim_{t\to\infty} \rho_\pi^{+(t)} = \rho_\pi^+$, and the induced Markov chain $M_\pi^+$ is ergodic.*

*Proof idea:* The detour to $s_{null}$ at the beginning of an admissible episode $\xi$ of length $n$ gives another admissible episode $\xi'$ of length $n+1$. As $\gcd(n+1, n) = 1$ for any positive integer $n$, the two episodes $\xi$ and $\xi'$ are co-prime in length, so by Lemma 2.1, $M_\pi^+$ is ergodic and has limiting distribution. See Appendix C.4 for the rigorous proof. $\qquad\square$

Moreover, the following theorem verifies that the perturbed episodic learning model preserves the decision-making related properties of the original model of the learning process.

**Theorem 3.** *In the $\epsilon$-perturbed model $\mathcal{M}^+$ of any episodic learning process $\mathcal{M}$, for every policy $\pi$, let $Q_\pi$ and $Q_\pi^+$ be the value functions of $\pi$ in $\mathcal{M}$ and $\mathcal{M}^+$, and let $\rho_\pi$ and $\rho_\pi^+$ be the limiting distribution under $\pi$ in $\mathcal{M}$ and $\mathcal{M}^+$, we have $Q_\pi = Q_\pi^+$ over $\mathcal{S} \times \mathcal{A}$, and $\rho_\pi \propto \rho_\pi^+$ over $\mathcal{S}$. Specifically,*

$$\rho_\pi(s) = \rho_\pi^+(s) / \left(1 - \rho_\pi^+(s_{null})\right) , \quad \forall s \in \mathcal{S}. \tag{5}$$

*Proof idea:* $Q_\pi^+$ remains the same because $\gamma(s_{null}) = 1$ and $R(s_{null}) = 0$, so $s_{null}$ is essentially transparent for any form of value assessment. Regarding $\rho_\pi^+$, we use a *coupling argument* to prove the proportionality. Intuitively we can simulate the rollout of $\mathcal{M}^+$ by inserting $s_{null}$ into a given trajectory of $\mathcal{M}$. This manipulation will certainly not change the empirical frequencies for states other

than $s_{null}$, so the proportionality between the *empirical* distributions is obvious. But the challenge is to connect the stationary distributions *behind* the empirical data – as $\mathcal{M}$ may not be ergodic, there is no guaranteed equivalence between its empirical and stationary distribution (the lack of such equivalence was our original motivation to introduce $\mathcal{M}^+$). So instead, we exploit the coupling effects between the two coupled trajectories to establish formal connection between the underlying steady states of $\mathcal{M}$ and $\mathcal{M}^+$. See Appendix C.5 for the complete proof. $\qquad\square$

The inserted state $s_{null}$ is intended to behave transparently in decision making while mixing the phases in empirical distribution at a cost of wasting at most one time step per episode. In fact, the proof of Theorem 3 suggests that we do not need to perturb the learning process in reality, but either a post-processing on given trajectories from the original model $\mathcal{M}$, or an on-the-fly *re-numbering* to the time index of states during the rollout of $\mathcal{M}$, would be enough to simulate the $\epsilon$-perturbation. With this perturbation trick in mind, we can now safely claim that *all* episodic learning processes that *we would consider* in RL are ergodic and admit limiting distributions – episodicity means ergodicity.

## 5    Conceptual implications

The ubiquity of steady state and ergodicity in episodic RL tasks may help connect concepts that are used in both episodic and continual RL yet were defined differently in the two RL formalisms.

For example, regarding performance measure, the episode-wise performance $J_{epi}$ was previously only well defined for episodic RL tasks, while the steady-state performance $J_{avg}$ was only well defined for continual RL tasks. The existence of unique stationary distribution $\rho_\pi$ in episodic learning process enables a rigorous connection between these two fundamental performance measures.

**Theorem 4.** *In any episodic learning process $\mathcal{M}$, for any policy $\pi$, $J_{epi}(\pi) = J_{avg}(\pi) \cdot \mathbb{E}_\pi[T]$. More explicitly,*

$$\mathop{\mathbb{E}}_{s\sim\rho_\pi}\left[R(s)\right] = \frac{\mathbb{E}_{\zeta\sim M_\pi}\left[\sum_{t=1}^{T(\zeta)} R(s_t)\right]}{\mathbb{E}_{\zeta\sim M_\pi}\left[T(\zeta)\right]} \tag{6}$$

*, where $T(\zeta) = \min\{t \geq 1 : s_t \in \mathcal{S}_\perp\}$ is as defined in Section 2.*

*Proof idea:* Averaging both sides of the Bellman equation (i.e. (2)) over distribution $\rho_\pi$, re-arranging a bit, then substituting in $\gamma(s) = \mathbb{1}(s \notin \mathcal{S}_\perp)$ will give the result. See Appendix C.6 for details. $\quad\square$

Note that the probability spaces at the two sides of (6) are different, one averaged over states, the other averaged over trajectories. Since in Theorem 4 the reward function $R$ can be arbitrary, the theorem gives a safe way to move between the two probability spaces for *any* function of state.

In particular, for arbitrary $s^*$ substituting $R(s) = \mathbb{1}(s = s^*)$ into (6) gives $\rho_\pi(s^*) = \mu_\pi^1(s^*)$, which immediately proves that the episode-wise visitation distribution $\mu_\pi^1$ (when well defined) is equal in values to the stationary distribution $\rho_\pi$. Interestingly, in continual RL, stationary distribution $\rho_\pi$ is used to formulate the concept of *on-policy distribution* which intuitively means "the data distribution of policy $\pi$", but the same term "on-policy distribution" is (re-)defined as the episode-wise visitation distribution $\mu_\pi^1$ in episodic RL [23]. In light of their equality now, we advocate to replace $\mu_\pi^1$ with $\rho_\pi$ as the definition of "on-policy distribution" in episodic RL (see Appendix D for more arguments).

Table 1 summarizes the discussions around conceptual unification so far in the paper. In this unified perspective, for both episodic and continual RL we work on the MDP model of the learning (instead of the decision) environment. On-policy distribution always means the unique stationary distribution of the policy. The performance measures remain unchanged, but they are now rigorously connected by a factor of $\mathbb{E}_\pi[T]$ (i.e. the average episode length under the policy $\pi$).

|  | Target Environment | On-policy Distribution | Performance Measure |
|---|---|---|---|
| Episodic | $\mathcal{M}_D$ (finite-horizon) | $\propto \mathbb{E}_\pi[\sum_{t\leq T} \mathbb{1}(s_t = s)]$ | $J_{epi}$ |
| Continual | $\mathcal{M}_L$ (assumed ergodic) | $\rho_\pi \ (= \lim_{t\to\infty} \rho_\pi^{(t)})$ | $J_{avg}$ |
| Episodic (*) | $\mathcal{M}_L$ (proved ergodic) | $\rho_\pi \ (= \lim_{t\to\infty} \rho_\pi^{(t)})$ | $J_{avg} \cdot \mathbb{E}_\pi[T] \ \ (= J_{epi})$ |

Table 1: Concept comparison between RL settings. Our account is summarized in the row with (*).

# 6 Algorithmic implications

Sample inefficiency and learning instability are two major challenges of RL practice at the moment. Being unified in the conceptual level, we can now borrow ideas from continual RL to efficiently collect stable training data for practical problems in episodic RL: Write the quantity we want to compute as an expectation over the stationary distribution of some policy $\beta$, perturb the learning environment if needed, rollout $\beta$ in the (perturbed) environment and wait for it to converge to the steady state, after which we obtain an unbiased sample point in every single time step. As a demonstration, the following subsection applies this idea to policy gradient algorithms.

## 6.1 Policy gradient

The availability of steady state in episodic tasks allows us to write the policy gradient of the episode-wise performance as a steady-state property of the corresponding policy.

**Theorem 5.** *(Steady-State Policy Gradient Theorem.) In episodic learning process $\mathcal{M}$, let $Q_\pi$ be the value function* (1) *using $\gamma(s) = \mathbb{1}(s \notin \mathcal{S}_\perp)$, then for any differentiable policy function $\pi(s, a; \theta)$,*

$$\nabla_\theta \, J_{epi}(\theta) = \left( \mathop{\mathbb{E}}_{\zeta \sim \pi(\theta)}[T] - 1 \right) \cdot \mathop{\mathbb{E}}_{s, a \sim \rho_\theta} \left[ Q_\theta(s, a) \, \nabla_\theta \log \pi(s, a; \theta) \, \Big| \, s \notin \mathcal{S}_\perp \right] \tag{7}$$

*where $s, a \sim \rho_\theta$ means $s \sim \rho_{\pi(\theta)}, a \sim \pi(s; \theta)$.*

*Proof idea:* Apply the $\nabla_\theta$ operator over both sides of the Bellman equation (2), then average both sides of the resulted equation over stationary distribution $\rho_\pi$ and re-arrange. Then substituting in $\gamma(s) = \mathbb{1}(s \notin \mathcal{S}_\perp)$ will give what we want. See the complete proof in Appendix C.7. □

Comparing with classic policy gradient theorems [24, 23] (i.e. with (3)), Theorem 5 establishes an *exact equality* directly from the gradient of the *end-to-end* performance $\nabla J_{epi}$ to an conditional expectation under the steady state of the target policy $\pi(\theta)$. This means that we can construct, from every non-terminal step $t$ after the convergence time, a (truly) unbiased estimate to the gradient of $\tilde{J}_{SSPG} \doteq J_{epi}$ using estimator $F_{SSPG}(s_t, a_t, \theta) = (\widehat{\mathbb{E}_\theta[T]} - 1) \cdot \widehat{Q_\theta} \cdot \nabla \log \pi(s_t, a_t; \theta)$. We can further reduce the variance of the estimator $F_{SSPG}$ by running $K$ parallel rollouts [13, 17], in which case the step data at time $t$ across all these independent rollouts form an i.i.d sample of size $K$.

Comparing with A3C [13], one of the best "online" policy gradient algorithms, the SSPG estimator as presented above differs in three places. First, A3C uses the discounted value functions $Q_\pi^{\gamma_c}$ (and a $V_\pi^{\gamma_c}$ baseline) with $\gamma_c < 1$ while SSPG uses $\gamma_c = 1$. Interestingly, while we already know that A3C does not compute the unbiased policy gradient because of this discounting [27], it is still less clear what it *computes instead* in episodic tasks ([27] only gave analysis in continual cases), but the proofs of Theorem 5 and 4 actually shed some light on the latter issue. See Appendix E for the details.

Secondly, in the SSPG estimator there is an additional term $\widehat{\mathbb{E}_\theta[T]} - 1$, which is the estimated Average Episode Length (*AEL* for short) of $\pi(\theta)$ minus 1. This AEL term was often treated as a "proportionality *constant*" and thus omitted in the literature [23]. However, Theorem 5, and specifically the subscript $\theta$ in $\mathbb{E}_\theta[T]$, makes it explicit that this term is not really a constant as the policy parameter $\theta$ is certainly changing in any effective learning. The corresponding average episode length can change by *orders of magnitude*, both increasingly (e.g. in Cart-Pole) and decreasingly (e.g. in Mountain-Car). While the dramatically changed AEL will not alter the gradient direction, it does affect *learning rate control* which is known to be of critical importance in RL practice.

We examined the impact of the changing AEL to the quality of policy gradient estimation in the Hopper environment of the RoboSchoolBullet benchmark [1]. We implemented two variants of SSPG, one using exactly $F_{SSPG}$, the other omitting the AEL term. Both used constant learning rate. In order to observe the potential bias, we run $K = 1,000,000$ parallel rollouts to reduce variance as much as possible, and used data from the single step at $\Delta t = 3 \cdot AEL$ across all rollouts to make each policy update. We then compared the estimated gradients from both SSPG variants with the ground-truth policy gradient at each update time (Figure 1 in appendix), and observed that the exact SSPG estimator indeed well aligns with the ground-truth gradients throughout the learning process, while the variant without the AEL term exhibits significant bias in its gradient estimates after 100 updates, due to a $7\times$ increase of AEL along the way. See F.1 for more experimental details.

Lastly, A3C works on the raw environments without perturbation and also does not wait for steady-state convergence. The next subsection is dedicated to the ramifications in this aspect.

## 6.2 Recursive perturbation

In the $\epsilon$-perturbation trick introduced in Section 4, an episode can only be perturbed for at most one step (and only at the beginning of the episode, with arbitrarily small probability $\epsilon$). The goal there is to enforce steady-state convergence while adding only *minimal* change to the raw environment. In practice, there is a simply way to "augment" the perturbation to facilitate *rapid* convergence, so that we don't need to wait too long before enjoying the efficiency of few-step sampling.

The key idea is that the same perturbation trick can be applied again to an already perturbed environment, which creates a detour visiting up to two null states at the beginning of an episode (and the "doubly perturbed" model must still remain ergodic). Repeatedly applying the same trick to the limit, we end up with a perturbed model with only one null state but the null state has a self-looping probability of $\epsilon$. The time steps that the agent will spend on the null state in an episode follows the geometric distribution with expectation $\epsilon/(1-\epsilon)$, so with $\epsilon$ approaching to 1, the perturbation effect can be arbitrarily large. We call this augmented perturbation method, *recursive perturbation*.

Interestingly, we found that empirically the recursive perturbation with $\epsilon = 1 - 1/\mathbb{E}[T]$ appears to enforce episodic learning processes to converge before $t^* = 3\,\mathbb{E}[T]$ across *various* environments. Specifically, we examined the time evolution of the marginal state distribution in three representative environments of RoboSchoolBullet [1]: Hopper, Humanoid, and HalfCheetah. Without perturbation, the raw environment of Hopper indeed converges around $t = 2 \cdot AEL$, but Humanoid converges only around $24 \cdot AEL$, which is more than $\times 10$ slower. Moreover, without perturbation the time evolution in HalfCheetah does not converge at all, but instead periodically fluctuates around its steady state forever. These observations suggest that perturbation is indeed *necessary* to assure convergence in real-world tasks. On the other hand, when recursive perturbation with $\epsilon = 1 - 1/\mathbb{E}[T]$ was applied, both Humanoid and HalfCheetah well approached to their respective steady states at $t = 3 \cdot AEL$.

To further test the generality of the "3-AEL convergence" phenomenon, we designed, as a worst-case scenario, a synthetic environment in which an episode goes through the state space $\mathcal{S} = \{1, \dots, n\}$ in a strictly round-robin manner (regardless of the actions). The rollout of such environment is fully deterministic, and probabilities of its state distribution $\rho^{(t)}$ fully concentrate at the single state $s = (t \mod n)$. We then applied the recursive perturbation with $\epsilon = 1 - 1/n$ to such environments with episode length ranging from $n = 20$ to 2000, and observed that for all of them, the marginal distribution $\rho^{(t^*)}$ at $t^* = 3n$ becomes *uniformly* distributed after the perturbation. Details about the perturbation experiments, on both RoboSchoolBullet and the synthetic one, can be found in F.2.

## 7 Discussions

Ergodicity is a phenomenon inherently linked to recurrence. In episodic tasks there are recognized recurrent states, the terminal states, thus its guaranteed ergodicity should not be too surprising with hindsight. Intriguingly, it turns out that in the end what becomes more suspicious is the ergodicity presumption in the continual setting, which requires the existence of some *implicit* recurrent states in the environment dynamic (see Proposition 1 and 2) – but if that were true, wouldn't it imply that an ergodic "continual task" is in fact episodic with respect to these implicit recurrent states (which serve as hidden terminal states of the task)? It then follows that a truly and purely continual task *cannot* be ergodic as usually postulated. Note that our definition of episodic learning process is not based on perceived category in the view of an observer, but based on intrinsic properties of the environment itself. In particular, the set of terminal states is defined as the equivalence closure of the set of initial states prescribed by $\rho_0$ (see Definition 1). It would be interesting to see if some "as-perceived" continual task is actually episodic learning process under a differently specified $\rho_0$.

The methodology presented in Section 6 essentially reduces the data collection problem of episodic RL to a Markov Chain Monte Carlo (MCMC) problem, so insights from the MCMC theory and practice could be potentially beneficial to RL too. In particular, more *mixing time analysis* and *non-equilibrium analysis* on episodic RL process can be another thread of interesting future works.

The idea of modeling the resetting procedure in episodic learning as normal steps in a multi-episode MDP has been mentioned in [25, 30, 32], which also used product-form value functions [25] to "block" the $\gamma$ from propagating across episode boundaries. Our paper used similar ideas, but made a forward move by identifying strong *structures* in the MDP thus constructed and by demonstrating both conceptual and pragmatic benefits of our theoretical insights. We believe the *learning* (instead of *decision*) environment as formulated by this highly-structured MDP deserves more research attention.

## Broader Impacts

This section discusses potential social impacts of this paper, as required by the NeurIPS 2020 program committee. This work mostly looked at a theoretical foundation of reinforcement learning, which, in the author's view, contributes to *understand* RL as a general and *natural* phenomenon of the world and of our society. At the engineering side, RL is in its nature an "online" learning paradigm that requires close interaction with the surround environment. This interaction as triggered along with the learning could possibly have unexpected effects especially if a less understood RL algorithm is deployed. The paper contains analysis on both theoretically justified and questionable aspects of existing RL practice. The author hopes such analysis can contribute to more informed and responsible decisions in using these approaches in practice. Finally, the perturbation trick and the new policy gradient estimator proposed in the paper may be integrated into fully-fledged RL algorithms in the future by others, which could in turn be used as ethically-neutral tools to influence the society, inevitably in both expected and unexpected ways.

## Acknowledgments

The author of the paper would like to thank his colleague Xu Wang for inspiring discussions on this work. The author also appreciates the many helpful comments from the anonymous reviewers of NeurIPS 2020 on earlier version of the paper. As funding disclosure, this research work received no project-dedicated financial support or engagement from any third party.

## Footnotes

[1]To simplify notations we will write $\rho_\theta, Q_\theta, V_\theta, J(\theta)$ for $\rho_{\pi(\theta)}, Q_{\pi(\theta)}, V_{\pi(\theta)}, J(\pi(\theta))$ from now on.

[2]Technically speaking, our approach is actually closer to an idea that [32] argues *against*; see A for details.

[3]Note that the default sample space, i.e. the set of all infinite rollout trajectories $\{\zeta\}$, is a uncountable set (even for *finite* state space $\mathcal{S}$), while the new sample space $\Xi_\pi$ is a countable set.

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
