[Supplementary Material]

# Appendices

## A   Notes on the MDP formulation

This section provides more discussions about the MDP formulation as introduced in Section 2. A faithful implementation of this formulation is included in our experimental code, which was used to generate all the results reported in Section 6 and Section F.

The MDP formulation assumes both $\mathcal{S}$ and $\mathcal{A}$ are countable sets. This is mainly for aligning to the standard markov chain theory, and also for enabling convenient notations like transition matrix $M$ and summation $\sum_{s,a}$. Results in this paper are readily generalizable to uncountable action spaces (which still induce countable transitions after marginalizing over the actions), and may also be generalized to uncountable state spaces based on the theory of *general-state-space* markov chains [11].

The MDP formulation also assumes state-based deterministic reward function $R$. While this formulation was used in many previous works [15], some literature [23] explicitly assign stochastic rewards to transitions, in the form of $R(r|s,a,s') = \mathbb{P}[r_{t+1} = r|s_t = s, a_t = a, s_{t+1} = s']$. Our reward-function formulation has no loss of generality here, as one can think of a "state" in our formulation as a $(s,r)$ pair in stochastic-reward models, with the transition-dependent stochasticity of the reward implicitly captured by the transition function $P$.

The MDP formulation has replaced the (often included) discounting constant $\gamma_c$ with the (often excluded) initial state distribution $\rho_0$. Similar perspective to "downgrade" the discounting constant was discussed in [23] (Chapter 10.4). As will become evident later in the paper, the discounting constant is neither necessary for the purpose of defining the problem, nor is it necessary for treating the problem. On the other hand, an explicit specification of $\rho_0$ is necessary to define the episode-wise performance $J_{epi}$ as used in all episodic tasks, and will be also needed to define the terminal states in the formulation of episodic learning process proposed in this paper.

A finite-horizon task can have degenerated steady states if it is formulated as an infinite-horizon MDP with an imaginary absorbing state $s_{\square}$. In the absorbing MDP formulation, any finite episode will end up with moving from its terminal state to the absorbing state $s_{\square}$, from there the rollout is trapped in the absorbing state forever without generating effective reward [23, 27]. As a result, the stationary (and limiting) distribution of the absorbing MDP of any finite-horizon task concentrates fully and only to the absorbing state, making it of limited use for designing and analyzing RL algorithms.

The definition of the steady-state performance measure $J_{avg}(\pi) \doteq \lim_{T\to\infty} \frac{1}{T}\sum_{t=1}^{T} R(s_t) = \mathbb{E}_{s\sim\rho_\pi}[R(s)]$, as introduced in Section 2, is based on the following *ergodic theorem* of markov chain.

**Proposition 3.** *In ergodic markov chain $M$ with stationary distribution $\rho_M$, let $f(s)$ be any function such that $\mathbb{E}_{s\sim\rho_M}[\,|f(s)|\,] < \infty$, then the time-average of $f$ converges almost surely to the state-average of $f$, i.e., $\lim_{T\to\infty} \mathbb{P}[\frac{1}{T}\sum_{t=1}^{T} f(s_t) = \mathbb{E}_{s\sim\rho_M}[f(s)]\,] = 1$. ([18], Theorem 74)*

The value function $Q$ is often specifically called the *action-value function*. We called the $Q$-function just value function because of its symmetric role with policy function $\pi$. In their classic form, the value function was often specifically defined as $Q_\pi^{\gamma_c}(s,a) \doteq \mathbb{E}_{s_1\sim P(s,a),\{s_{\geq 2}\}\sim M_\pi}[\sum_{t=1}^{\infty} R(s_t)\cdot\gamma_c^{t-1}]$, in which the constant $0 \leq \gamma_c < 1$ is called the *discounting factor*. The specialized definition of $Q_\pi^{\gamma_c}$ entails the specialized version of Bellman equation $Q_\pi^{\gamma_c}(s,a) = \mathbb{E}_{s'\sim P(s,a),a'\sim\pi(s')}[R(s') + \gamma_c\cdot Q_\pi^{\gamma_c}(s',a')]$, and induces the specialized version of state-value function $V_\pi^{\gamma_c}(s) \doteq \mathbb{E}_{a\sim\pi(s)}[Q_\pi^{\gamma_c}(s,a)]$.

The product of $\gamma$ in the general-form value functions (1) was originally proposed as a virtual probabilistic termination in the agent's mind [22], but was latter found useful to account for the real episode-boundaries when the MDP is used to model multi-episode rollouts [30, 32]. Our paper uses product-form value functions for the same purpose as in [30] and [32]. However, the value functions in our treatment use state-based discounting, which is probably closer to the method of Figure 1(c) in [32], and was considered an sub-optimal design in that paper. In fact, [32] attributes much of its main technical contribution to the adoption and analysis of transition-based discounting, and concludes that "*transition-based discounting is necessary to enable the unified specification of episodic and continuing tasks*" (see [32] Section 2.1, Section 6, and Section B). Nevertheless, despite this major technical disparity, we think the approach as described in Section 3 of our paper actually aligns with [30] and [32] in terms of the bigger idea that connecting finite episodes into a single infinite-horizon MDP greatly helps with unifying the episodic and continual formalisms.

## B   Gradient estimation in RL algorithms

As mentioned in Section 2, many RL algorithms can be interpreted under a common and basic idea that seeks to find a parameterized policy function $\pi(\theta)$, a surrogate objective $\tilde{J}$, and an estimator function $F$, such that

good policies with respect to the end-to-end performance measure (typically $J_{avg}$ in continual tasks and $J_{epi}$ in episodic tasks) can be found by optimizing the surrogate objective $\tilde{J}$ via stochastic gradient methods, where $\nabla_\theta \tilde{J}(\theta_t)$ is estimated by averaging $F$ over some data set $\mathcal{D}_t$, that is

$$\nabla_\theta \tilde{J}(\theta_t) \approx \frac{1}{|\mathcal{D}_t|} \sum_{(s,a) \in \mathcal{D}_t} F(s, a, \theta_t). \tag{8}$$

RL algorithms (policy-based or value-based, on-policy or off-policy) differ in how its policy function $\pi$ is parameterized, as well as in how $\tilde{J}$, $F$, and $\mathcal{D}$, are constructed. This section briefly reviews the specific forms of $\tilde{J}$ and $F$, as well as the prescribed distribution behind $\mathcal{D}_t$, that are used in some popular RL algorithms at the moment. To simplify notations we write $\rho_\theta$, $Q_\theta$, $V_\theta$, $J(\theta)$ for $\rho_{\pi(\theta)}$, $Q_{\pi(\theta)}$, $V_{\pi(\theta)}$, $J(\pi(\theta))$, respectively, from now on. As all algorithms discussed in this section use constant discounting, we write $\gamma$ (instead of $\gamma_c$) for the discounting constant (only) in this section.

In the *policy-based approach*[33, 24, 8, 19, 15, 17], the policy $\pi$ is a function directly differentiable to policy parameter $\theta$ (that is, $\nabla_\theta \pi$ is computable). In its most classic and popular form, the surrogate objective $\tilde{J}$ is chosen to be a *discounted episodic return* $J_{epi}^\gamma(\theta) \doteq \mathbb{E}_{s_0 \sim \rho_0}[V_\theta^\gamma(s_0)]$, whose estimator $F$ is derived as [24]

$$\nabla J_{epi}(\theta) \approx \nabla J_{epi}^\gamma(\theta) = \sum_{t=0}^\infty \gamma^t \cdot \mathbb{E}_{s_t \sim \rho_\theta^{(t)}} \mathbb{E}_{a_t \sim \pi(s_t; \theta)} \left[ \nabla \log \pi(s_t, a_t; \theta) \cdot Q_\theta^\gamma(s_t, a_t) \right] \tag{9}$$

$$= \frac{1}{1 - \gamma} \cdot \mathbb{E}_{s \sim \mu_\theta^\gamma} \mathbb{E}_{a \sim \pi(s; \theta)} \left[ \nabla \log \pi(s_t, a_t; \theta) \cdot Q_\theta^\gamma(s, a) \right] \tag{10}$$

$$\approx \frac{1}{|\mathcal{D}|} \sum_{(s_i, a_i) \in \mathcal{D}} \nabla \log \pi(s_t, a_t; \theta) \cdot \widehat{Q_\theta^\gamma}(s_i, a_i). \tag{11}$$

In (10), $\mu_\theta^\gamma \doteq \sum_{t=0}^\infty \rho_\theta^{(t)} \cdot \gamma^t (1 - \gamma)$ is sometimes called the *(normalized) discounted visitation distribution*, and $\widehat{Q_\theta^\gamma}(s_i, a_i)$ is some approximation of the value $Q_\theta^\gamma(s_i, a_i)$. In above, $F_{pg}(s, a, \theta) = \nabla \log \pi(s, a; \theta) \cdot \widehat{Q_\theta^\gamma}(s, a)$.

The vanilla REINFORCE algorithm [33, 23] uses one-step data $(s_t, a_t)$ from a single rollout to construct the data set $\mathcal{D}$ for each policy update. Modern variants of it employ batch-mode updates [15, 4], using data accumulated from multiple episodes to construct the data set $\mathcal{D}$. The A3C algorithm [13] uses the same surrogate objective $J_{epi}^\gamma$ and the same estimator $F_{pg}$, but constructs $\mathcal{D}$ using data from a small time window (e.g. five consecutive steps[13]) of multiple *parallel and independent* rollouts. The PPO algorithm [17] collects data set $\mathcal{D}$ in similar way, but conduct multiple policy updates on a single data set $\mathcal{D}$, thus improving sample efficiency. To keep the policy updates well directed, PPO uses a slightly different surrogate objective that majorizes $J_{epi}^\gamma$ around the base parameter $\theta^{old}$, an idea first employed in the TRPO algorithm [15]. In all these RL algorithms, the data set $\mathcal{D}$ follows the on-policy distribution of the *target policy* $\pi(\theta^{old})$, and are thus called *on-policy algorithms*.

In the *value-based approach* [31, 12, 29, 6, 9, 5, 7], the agent policy $\pi$ is parameterized indirectly *through* a differentiable function $Q$. For example, $\pi$ may be a greedy policy that has zero selection-probability for all sub-optimal actions with respect to $Q(\theta)$, i.e., with $\pi(s, a; \theta) = 0$ for $a \notin \arg\max_{\bar{a}} Q(s, \bar{a}; \theta)$. In this case $\nabla_\theta \pi$ is generally not computable, but $\nabla_\theta Q$ is. In the most classic form of this approach, the surrogate objective $\tilde{J}$ is chosen to be the so-called *Projected Bellman Error* $J_{PBE}(\theta) \doteq \sum_{s \in \mathcal{S}} \sum_{a \in \mathcal{A}} \delta^2(s, a; \theta)$, where $\delta(s, a; \theta) \doteq Q(s, a; \theta) - \mathbb{E}_{s' \sim P(s,a), a' \sim \pi(s'; \theta^{old})}[R(s') + \gamma Q(s', a'; \theta^{old})]$, whose estimator is derived as [12]

$$\nabla J_{PBE}(\theta) \approx \mathbb{E}_{s,a \sim \mathcal{U}(\mathcal{S} \times \mathcal{A})}[\nabla Q(s, a; \theta) \cdot \delta(s, a; \theta)] \tag{12}$$

$$\approx \frac{1}{|\mathcal{D}|} \sum_{(s_i, a_i) \in \mathcal{D}} \nabla Q(s_i, a_i; \theta) \cdot \widehat{\delta}(s_i, a_i; \theta), \tag{13}$$

where $\mathcal{U}(\mathcal{S} \times \mathcal{A})$ can be any positive distribution over the states and actions, $\widehat{\delta}(s_i, a_i; \theta)$ is some approximation of $\delta(s_i, a_i; \theta)$, and $F_{PBE}(s, a, \theta) = \nabla Q(s, a; \theta) \cdot \widehat{\delta}(s, a; \theta)$.

Similar to the case in policy-based approach, early value-based algorithms such as Q-Learning [31] used one-step data $(s_t, a_t)$ from a single rollout to construct the data set $\mathcal{D}$ for each policy update based on (13), while modern variants of it typically conduct batch-mode updates, again either using multiple-episode data from a single rollout [12] or using data of small time window from parallel rollouts [13]. The basic surrogate objective $J_{PBE}$ and its estimator $F_{PBE}$ used in (13) can also be improved in many ways, such as using two (weakly- or un-correlated) base parameters $\theta^{old}$ in the $\delta$ function [29, 5], and adding entropy-regularization terms [16, 7]. Variants of (13) that are applicable to continuous action spaces were also proposed [9, 6].

In order to comprehensively approximate the positive distribution $\mathcal{U}(\mathcal{S} \times \mathcal{A})$ in (12), these value-based algorithms typically employ some behavior policy $\beta$ that is more exploratory than the target policy $\pi$, so that the data set $\mathcal{D}$ in (13) follows the on-policy distribution $\rho_\beta$. To improve sample efficiency and reduce auto-correlation,

the sample set $\mathcal{D}$ used by modern value-based algorithms is usually a mixture of data from multiple behavior policies $\beta_t$'s that the agent has been using over time $t$ [12], in which case $\mathcal{D}$ follows a mixture of the on-policy distributions $\rho_{\beta_t}$. As the data set $\mathcal{D}$ does not follow the on-policy distribution of the target policy $\pi$ (or of any single policy) in this case, algorithms based on such data set $\mathcal{D}$ is called *off-policy algorithms*.

As we can see, all the RL algorithms discussed above, policy-based or value-based, on-policy or off-policy, they all rely on the capability to obtain high-quality data from rollouts (sequential or parallel) that follows a desired distribution. Our work mainly concern about the theoretical underpins and proper sampling strategies to generate the data as required, and is thus complementary to most of the works reviewed above.

# C   Complete Proofs

## C.1   Lemma 1.1

*Proof.*  Due to the homogeneity condition of episodic learning, a reachable state must be reachable in one episode and from any terminal state. So let $\xi$ be such an admissible episode trajectory under $\pi$ that go through the state $s$, and suppose $\xi$ terminates at an arbitrary terminal state $s^*$, there must be an admissible episode trajectory $\xi'$ that starts from $s^*$ and go through the state $s'$. Due to the finiteness condition of episodic learning, both $\xi$ and $\xi'$ take finite steps, so the concatenated trajectory $\xi\xi'$ contains a finite path $s \to s'$ as subsequence.  $\square$

## C.2   Lemma 1.2

*Proof.*  For the purpose of the proof, suppose we roll out $M_\pi$ starting from an arbitrary $s$, so $T_s$ is the time index of the first reccurrence of $s$ in such a rollout. To make the proof rigorous, we first slightly re-formulate the probability space of $\mathbb{E}[T_s]$: Imagine again we rollout $M_\pi$ starting from $s$ but we terminate the rollout immediately after the rollout returns to $s$ for the first time. The sample space of such truncated rollout, denoted by $\Omega_s$, consists of all the finite trajectories $\xi = (s, \ldots, s)$ in which $s$ shows up *only* at the first and the last time step. Let $T_s(\xi)$ denote the recurrence time of $s$ (i.e. the last time step) in a specific $\xi \in \Omega_s$, the probability to obtain such a $\xi$ from the truncated rollout is $\mathbb{P}_\xi[\xi] = \prod_{t \geq 1} M_\pi(s_{t-1}, s_t)$, and $\sum_{\xi \in \Omega_s} \mathbb{P}_\xi[\xi] = 1$ due to the finiteness condition of episodic learning process. The expected recurrence time in the truncated rollout is the same as the one in the full rollout as in the former case we only truncate *after* the recurrence, so $\mathbb{E}_{\zeta \sim M_\pi}[T_s] = \mathbb{E}_{\xi \in \Omega_s}[T_s]$. In the rest of this proof, when we write $\mathbb{E}[T_s]$ we mean $\mathbb{E}_{\xi \in \Omega_s}[T_s]$.

Let $n_s$ be the number of episodes *completed* before time $T_s$ (i.e. by time step $T_s - 1$), we have $\mathbb{E}[T_s] = \sum_{k \geq 0} \mathbb{P}[n_s = k] \cdot \mathbb{E}[T_s | n_s = k]$.

$n_s = k$ means that the first recurrence of $s$ occurs in the $(n_s + 1)$-th episode. Due to the homogeneity condition, there is a *uniform* episode-wise hitting probability $\mathbb{P}_\xi[s \in \xi]$ which applies to all the $n_s + 1$ episodes. Denoting $\alpha_s = \mathbb{P}_\xi[s \in \xi]$, we have $\mathbb{P}[n_s = k] = (1 - \alpha_s)^k \alpha_s$.

On the other hand, as the recurrence time $T_s$ falls in the $n_s + 1$-th episode, it must be upper bounded by the sum of lengths of all the $n_s + 1$ episodes. Thus, due to the finiteness condition, we have $\mathbb{E}[T_s | n_s = k] \leq (k + 1) \cdot \mathbb{E}_\xi |\xi| < +\infty$.

Putting things together, we have $\mathbb{E}[T_s] \leq \sum_{k \geq 0} (1 - \alpha_s)^k \alpha_s \cdot (k + 1) \mathbb{E}_\xi |\xi| = \alpha_s \mathbb{E}_\xi |\xi| \cdot \sum_{k \geq 0} (1 - \alpha_s)^k (k + 1) = \alpha_s \mathbb{E}_\xi |\xi| \cdot \frac{1}{\alpha_s^2} = \mathbb{E}_\xi |\xi| / \alpha_s$. Note that $\alpha_s > 0$ because $s$ is reachable in $M_\pi$.  $\square$

## C.3   Lemma 2.1

*Proof.*  Lemma 1.1 shows that $M_\pi$ is irreducible over $\mathcal{S}_\pi$, so we only need to identify one *aperiodic* state $s \in \mathcal{S}_\pi$, which will prove that $M_\pi$ is ergodic, then by Proposition 2 the stationary distribution $\rho_\pi$ is also the limiting distribution over $\mathcal{S}_\pi$ (and clearly $\lim_{t \to \infty} \rho_\pi^{(t)} = 0 = \rho_\pi$ for unreachable $s \notin \mathcal{S}_\pi$).

Consider episodes in $M_\pi$ that end at some terminal state $s_1 \in \mathcal{S}_\perp$ after $n$ steps. Such an episode can start from any terminal state, including $s_1$ itself. Let $\xi_{1,1}$ be the trajectory of such an episode, which is thus a $n$-step recurrence of state $s_1$. On the other hand, due to the assumed condition in the lemma, for some such $n$ we can find episodes with co-prime length $m$ with $\gcd(n, m) = 1$. Let $\xi_{1,2}$ be such an episode trajectory of length $m$, which starts again at $s_1$ but ends at some terminal state $s_2 \in \mathcal{S}_\perp$.

Now if $s_1 = s_2$, then $\xi_{1,2}$ is another recurrence trajectory of $s_1$ which has co-prime length with $\xi_{1,1}$, thus $s_1$ is aperiodic. Otherwise if $s_1$ and $s_2$ are different terminal states, then we replace the initial state of $\xi_{1,1}$ from $s_1$ to $s_2$, obtaining a third episode trajectory $\xi_{2,1}$, which starts from $s_2$ and ends at $s_1$ after $|\xi_{2,1}| = |\xi_{1,1}| = n$ steps. Consider the concatenated trajectory $\xi_{1,2}\xi_{2,1}$, which is the trajectory of two consecutive episodes that first goes from $s_1$ to $s_2$ in $m$ steps, then goes from $s_2$ back to $s_1$ in $n$ steps, thus form a $(m + n)$-step recurrence of $s_1$.

As $\gcd(m+n, n) = \gcd(m, n) = 1$ for any $n \neq m$, we know $\xi_{1,1}$ and $\xi_{1,2}\xi_{2,1}$ are two recurrences of $s_1$ with co-prime lengths, thus $s_1$ is still aperiodic. □

## C.4  Theorem 2

*Proof.* Consider an arbitrary admissible episode $\xi$ in the Markov chain $M_\pi$ induced by $\pi$ in the original learning model $\mathcal{M}$. Let the episode length $|\xi| = n$ be an arbitrary integer $n > 0$. $\xi$ is still a possible episode under $\pi$ in the perturbed model $\mathcal{M}^+$. In particular, from the definition of $P^+$ (in Definition 2) we have $\mathbb{P}_{\xi \sim M_\pi^+}[\xi] = (1-\epsilon) \cdot \mathbb{P}_{\xi \sim M_\pi}[\xi]$.

On the other hand, in the perturbed model, the trajectory $\xi = (s_0, s_1, \ldots, s_n)$ is accompanied with a "detoured" trajectory $\xi' = (s_0, s_{null}, s_1, \ldots, s_n)$, which is the same with $\xi$ except for the detour steps $s_0 \to s_{null} \to s_1$. The detour is always possible under any policy $\pi$ as both $P^+(s_{null}|s_0, a) = \epsilon$ and $P^+(s_1|s_{null}, a) = P(s_1|s_0, a)$ are action-agnostic and non-zero. So, $M_\pi$ admits both $\xi$ and $\xi'$, which have episode lengths $n$ and $n+1$, respectively. As $\gcd(n+1, n) = 1$ for any positive integer $n$, we got two episodes with co-prime lengths now, then from Lemma 2.1 we know $M_\pi^+$ is ergodic and thus has limiting distribution. □

## C.5  Theorem 3

*Proof.* As terminal states have $\gamma = 0$, the calculation of the $\prod \gamma_\tau$ term in $Q_\pi^+(s, a)$ (in Eq. (1)) truncates at the end of an episode. As the auxiliary state $s_{null}$ is reachable only from a terminal state in the very first step of an episode, for any other $s \notin \mathcal{S}_\perp$ the whole transition model remains the same within an episode, thus $Q_\pi^+(s, a) = Q_\pi(s, a)$ for $s \notin \mathcal{S}_\perp$. For $s \in \mathcal{S}_\perp$, their action values is also unchanged because, with $\gamma(s_{null}) = 1$ and $R(s_{null}) = 0$, the detour to $s_{null}$ does not lead to any discounting nor any addition reward. The only effect is the prolonged episode lengths.

On the other hand, we use a coupling argument to prove $\rho_\pi^+ \propto \rho_\pi$ over $\mathcal{S}$. Consider the *coupled sampling* procedure shown in Algorithm 1. For ease of notation we use "null" to denote the auxiliary state $s_{null}$ in the rest of the proof. Consider the status of the variables in the procedure at an arbitrary time $t > 0$. $s_t$ is simply a regular sample of the original model $M_\pi$, so $s_t \sim \rho_\pi^{(t)}$.

$\zeta^+$ is obtained by inserting with probability $\epsilon$ a null state after each terminal state in the original rollout trajectory $\zeta = \{s_t\}$. Comparing with Definition 2, we see that $\zeta^+$ follows the perturbed model $(M_\pi^+, \rho_0)$ under $\pi$. More accurately, let $s_t^+$ denote the state in $\zeta^+$ at time $t$, we have $s_t^+ \sim \rho_\pi^{+(t)}$.

$z_t$ always equals an old state that $\zeta$ has encountered at an earlier time, with $\Delta_t$ being the time difference, so $z_t$ (as a random variable well defined by Algorithm 1) must follow the same marginal distribution with $s_{t-\Delta_t}$, thus we have $z_t \sim \rho_\pi^{(t-\Delta_t)}$.

---

**Algorithm 1:** a coupled sampling procedure

**Input:** $M_\pi$, $\rho_0$, an i.i.d. sampler $random() \sim \mathcal{U}[0,1]$
**Output:** an infinite tajectory $(z_0, z_1 \ldots)$

1  sample $s_0 \sim \rho_0$
2  initialize trajectory $\zeta^+ \leftarrow (s_0)$
3  set $\Delta_0 \leftarrow 0$
4  set $z_0 \leftarrow s_0$
5  **for** $t = 1, 2, \ldots$ **do**
6      sample $s_t \sim M_\pi(s_{t-1})$
7      **if** $s_{t-1}^+ \in \mathcal{S}_\perp$ **and** $random() < \epsilon$ **then**
8          └ append $(s_{null}, s_t)$ to $\zeta^+$
9      **else**
10         └ append $s_t$ to $\zeta^+$
11     set $\Delta_t \leftarrow \#s_{null}$ in subsequence $\zeta_{0:t-1}^+$
12     set $z_t \leftarrow s_{t-\Delta_t}$

| $\zeta$ | $\zeta^+$ | $\Delta_t$ | $\{z_t\}$ |
|---------|-----------|------------|-----------|
| $s_0$ | $s_0$ | 0 | $s_0$ |
| $s_1$ | null | 0 | $s_1$ |
| $s_2$ | $s_1$ | 1 | $s_1$ |
| $s_3$ | $s_2$ | 1 | $s_2$ |
| $s_4$ | null | 1 | $s_3$ |
| $s_5$ | $s_3$ | 2 | $s_3$ |
|  | $s_4$ |  |  |
|  | $s_5$ |  |  |

An example running of Algorithm 1. The table shows a snapshot when $t = 5$. In the $\zeta^+$ column, "null" denotes $s_{null}$, and terminal states ($s_0, s_1, s_2, s_4$) are highlighted with gray background.

---

In above we obtained the marginals of each of the three random trajectories maintained in Algorithm 1, next we consider the coupling effects between them. Observe that when $s_t^+$ is not null, it is (also) an old state in $\zeta$ with time index shifted by the number of null states inserted before $t$ (i.e. by $t-1$) in previous samplings, which is *the* state the procedure uses to assign value for $z_t$. In other words, $z_t = s_t^+$ conditioned on $s_t^+ \neq$ null. Therefore, for any state $s \in \mathcal{S}$,

$$\mathbb{P}[z_t = s] = \mathbb{P}[s_t^+ = s | s_t^+ \neq \text{null}] = \frac{\mathbb{P}[s_t^+ = s] \cdot \mathbb{P}[s_t^+ = \text{null} | s_t^+ = s]}{\mathbb{P}[s_t^+ \neq \text{null}]} = \frac{\mathbb{P}[s_t^+ = s] \cdot 1}{\mathbb{P}[s_t^+ \neq \text{null}]}. \quad (14)$$

(14) holds for any $t > 0$, thus must also hold at limit when $t \to \infty$. As $s_t^+$ follows the perturbed Markov chain $M_\pi^+$, it is known to have limiting distribution as proved in Theorem 2, thus the limits of both $\mathbb{P}[s^+ = s]$ and $\mathbb{P}[s_t^+ \neq \text{null}]$ at the rhs of (14) exist, which means the limit $\lim_{t \to \infty} \mathbb{P}[z_t = s]$ of the lhs must also exists. Let $c_\pi = \lim_{t \to \infty} \mathbb{P}[s_t^+ \neq \text{null}]$, we have

$$\lim_{t \to \infty} \mathbb{P}[z_t = s] = \lim_{t \to \infty} \mathbb{P}[s_t^+ = s]/c_\pi = \rho_\pi^+(s)/c_\pi. \tag{15}$$

Note that $c_\pi > 0$ because $\epsilon < 1$ by definition.

Now we only need to prove $\lim_{t \to \infty} \mathbb{P}[z_t = s] = \rho_\pi$. For that purpose, first observe that $\Delta$ gets increased in Algorithm 1 only if $\zeta^+$ entered $s_{null}$ in the last step – only in that case $\Delta$ has a new "null" counted in. In other words, $\Delta_{t+1} = \Delta_t + 1$ if $s_t^+ = \text{null}$, otherwise $\Delta_{t+1} = \Delta_t$.

Now consider the value of $z_{t+1}$. When $s_t^+ = \text{null}$, we have $z_{t+1} = s_{t+1-\Delta_{t+1}} = s_{t+1-\Delta_t-1} = s_{t-\Delta_t}$. When $s_t^+ \neq \text{null}$, we have $z_{t+1} = s_{t+1-\Delta_{t+1}} = s_{t+1-\Delta_t}$. Let $c_\pi^{(t)} = \mathbb{P}[s_t^+ \neq \text{null}]$ for any $t$, then for any $s \in \mathcal{S}$ we have

$$\mathbb{P}[z_{t+1} = s] = (1 - c_\pi^{(t)}) \cdot \mathbb{P}[s_{t-\Delta_t} = s] + c_\pi^{(t)} \cdot \mathbb{P}[s_{t+1-\Delta_t} = s]. \tag{16}$$

Due to (15), the two terms $\mathbb{P}[z_{t+1} = s]$ and $\mathbb{P}[s_{t-\Delta_t} = s] = \mathbb{P}[z_t = s]$ in (16) have the same limit. Taking both sides of (16) to limit and re-arranging, yields

$$c_\pi \cdot \lim_{t \to \infty} \mathbb{P}[s_{t+1-\Delta_t} = s] = \lim_{t \to \infty} \mathbb{P}[z_{t+1} = s] - (1 - c_\pi) \lim_{t \to \infty} \mathbb{P}[z_t = s] = c_\pi \cdot \lim_{t \to \infty} \mathbb{P}[s_{t-\Delta_t} = s]. \tag{17}$$

The two ends of (17) gives $\lim_{t \to \infty} \mathbb{P}[s_{t+1-\Delta_t} = s] = \lim_{t \to \infty} \mathbb{P}[s_{t-\Delta_t} = s]$, which holds for all states $s \in \mathcal{S}$, thus by definition of the marginal state distribution we have

$$\lim_{t \to \infty} \rho_\pi^{(t-\Delta_t)} = \lim_{t \to \infty} \rho_\pi^{(t+1-\Delta_t)} = \lim_{t \to \infty} \rho_\pi^{(t-\Delta_t)} \cdot M_\pi. \tag{18}$$

The two ends of (18) gives a fixed point of the operator $M_\pi$, for which $\rho_\pi$ is known to be the only solution, so $\lim_{t \to \infty} \rho_\pi^{(t-\Delta_t)} = \rho_\pi$. Further combining with (15), we finally obtain

$$\rho_\pi(s) = \lim_{t \to \infty} \mathbb{P}[s_{t-\Delta_t} = s] = \lim_{t \to \infty} \mathbb{P}[z_t = s] = \rho_\pi^+(s)/c_\pi, \tag{19}$$

for every $s \in \mathcal{S}$. $\qquad \square$

## C.6 Theorem 4

*Proof idea:* Averaging both sides of the Bellman equation (2) over the stationary distribution $\rho_\pi$ and re-arranging a bit, will give

$$\mathop{\mathbb{E}}_{s \sim \rho_\pi} \left[ R(s) \right] = \mathop{\mathbb{E}}_{s \sim \rho_\pi} \mathop{\mathbb{E}}_{a \sim \pi(s)} \left[ \left( 1 - \gamma(s) \right) \cdot Q_\pi(s, a) \right]. \tag{20}$$

Then substituting $\gamma(s) = \mathbb{1}(s \notin \mathcal{S}_\perp)$ into (20) will cancel out all the terms corresponding to non-terminal states, leaving only $V_\pi(s_\perp)$ at the RHS. See the complete proof in Appendix C.6. $\qquad \square$

*Proof.* Averaging the both sides of the Bellman equation (2) over the stationary distribution $\rho_p i$ and re-arranging, yields

$$
\begin{aligned}
0 &= \mathop{\mathbb{E}}_{\substack{s \sim \rho_\pi \\ a \sim \pi(s)}} \left[ \mathop{\mathbb{E}}_{\substack{s' \sim P(s,a) \\ a' \sim \pi(s')}} [R(s') + \gamma(s')Q_\pi(s', a')] - Q_\pi(s, a) \right] \\
&= \mathop{\mathbb{E}}_{\substack{s \sim \rho_\pi \\ a \sim \pi(s)}} \mathop{\mathbb{E}}_{\substack{s' \sim P(s,a) \\ a' \sim \pi(s')}} \left[ R(s') \right] + \mathop{\mathbb{E}}_{\substack{s \sim \rho_\pi \\ a \sim \pi(s)}} \mathop{\mathbb{E}}_{\substack{s' \sim P(s,a) \\ a' \sim \pi(s')}} \left[ \gamma(s')Q_\pi(s', a') \right] - \mathop{\mathbb{E}}_{\substack{s \sim \rho_\pi \\ a \sim \pi(s)}} \left[ Q_\pi(s, a) \right] \\
&= \mathop{\mathbb{E}}_{s \sim \rho_\pi} [R(s)] + \mathop{\mathbb{E}}_{\substack{s \sim \rho_\pi \\ a \sim \pi(s)}} [\gamma(s)Q_\pi(s, a)] - \mathop{\mathbb{E}}_{\substack{s \sim \rho_\pi \\ a \sim \pi(s)}} [Q_\pi(s, a)] \\
&= \mathop{\mathbb{E}}_{s \sim \rho_\pi} [R(s)] - \mathop{\mathbb{E}}_{\substack{s \sim \rho_\pi \\ a \sim \pi(s)}} [(1 - \gamma(s))Q_\pi(s, a)]
\end{aligned}
\tag{21}
$$

Substituting $\gamma(s) = \mathbb{1}(s \notin \mathcal{S}_\perp)$ into (21) will remove all terms corresponding to non-terminal states, giving

$$
\begin{aligned}
\mathop{\mathbb{E}}_{s \sim \rho_\pi} [R(s)] &= \sum_{s \in \mathcal{S}_\perp} \rho_\pi(s) \cdot \mathop{\mathbb{E}}_{a \sim \pi(s)} [Q_\pi(s, a)] \\
&= \left( \sum_{s \in \mathcal{S}_\perp} \rho_\pi(s) \right) \cdot V_\pi(s_\perp) = \left( \sum_{s \in \mathcal{S}_\perp} \rho_\pi(s) \right) \cdot J_{epi}
\end{aligned}
\tag{22}
$$

Then the following proposition turns $\sum_{s \in \mathcal{S}_\perp} \rho_\pi(s)$ into $1/\mathbb{E}_{\zeta \sim M_\pi}[T(\zeta)]$ as desired.

**Proposition 4.** $\mathbb{E}_{\zeta \sim M_\pi}[T(\zeta)] = 1/\sum_{s \in \mathcal{S}_\perp} \rho_\pi(s)$

*Proof.* Consider the MDP $\mathcal{M}'$ obtained by grouping all terminal states in $\mathcal{M}$ into a single "macro-state" $s_\perp$. Note that in general, grouping terminal states into one will change how the model delivers the rewarding feedback to the agent, as $R(s)$ may vary between terminal states. However, for the purpose of this proof, the transition dynamics of $\mathcal{M}$ and $\mathcal{M}'$ under *given* policy $\pi$ (that is, the markov chains $M_\pi$ and $M'_\pi$) will remain the same as all terminal states are homogeneous in transition probabilities. In particular, the expected episode length of $M_\pi$ will be the same with that of $M'_\pi$, from which we obtain

$$
\begin{aligned}
\mathbb{E}_{\zeta \sim M_\pi}[T(\zeta)] &= \mathbb{E}_{\zeta \sim M'_\pi}[T(\zeta)] = \mathbb{E}_{\zeta \sim M'_\pi}[T_{s_\perp}(\zeta)] \\
&= 1/\rho'(s_\perp) = 1/(1 - \sum_{s \in \mathcal{S} \setminus \mathcal{S}_\perp} \rho'(s)) \\
&= 1/(1 - \sum_{s \in \mathcal{S} \setminus \mathcal{S}_\perp} \rho(s)) = 1/\sum_{s \in \mathcal{S}_\perp} \rho_\pi(s).
\end{aligned}
\tag{23}
$$

In above $\mathbb{E}[T] = \mathbb{E}[T_{s_\perp}]$ because $s_\perp$ is the only terminal states in $M'_\pi$, and $\rho'(s)$ exists because $\mathcal{M}'$ is episodic. □

Substituting Proposition 4 back to (22) completes the proof. □

## C.7 Theorem 5

*Proof idea:* Adding the $\nabla_\theta$ operator over both sides of the Bellman equation (2), then averaging both sides (including $\nabla$) over stationary distribution $\rho_\pi$, and re-arranging a bit, will give

$$
\mathbb{E}_{s,a \sim \rho_\theta}\left[\left(1 - \gamma(s)\right)\nabla_\theta Q_\theta(s,a)\right] = \mathbb{E}_{s,a \sim \rho_\theta}\left[\gamma(s)\, Q_\theta(s,a)\nabla_\theta \log \pi(s,a;\theta)\right].
\tag{24}
$$

Then substituting $\gamma(s) = \mathbb{1}(s \notin \mathcal{S}_\perp)$ into (24) will leave only terminal states at LHS while leaving only non-terminal states at RHS. For terminal states, $\nabla Q_\theta(s_\perp, a) = \nabla J_{epi}$, which brings in the objective. Further re-arranging will give what we want. See the complete proof in Appendix C.7. □

*Proof.* First consider the quantity $\mathbb{E}_{s,a \sim \rho_\theta}\left[\nabla_\theta Q_\theta(s,a)\right]$ in its general form (not necessarily with the specific $\gamma$ function as assumed), we have

$$
\begin{aligned}
\mathbb{E}_{s,a \sim \rho_\theta}\left[\nabla_\theta Q_\theta(s,a)\right] &= \mathbb{E}_{s,a \sim \rho_\theta}\left[\nabla_\theta \mathbb{E}_{s' \sim P(s,a)}\left[R(s') + \gamma(s')\sum_{a'} \pi(a'|s';\theta)\, Q_\theta(s',a')\right]\right] \\
&= \mathbb{E}_{s,a \sim \rho_\theta}\mathbb{E}_{s' \sim P(s,a)}\left[\gamma(s')\sum_{a'}\nabla_\theta\left(\pi(a'|s';\theta)\, Q_\theta(s',a')\right)\right] \\
&= \mathbb{E}_{s \sim \rho_\theta}\left[\gamma(s)\sum_{a}\left(Q_\theta(s,a)\nabla_\theta\pi(a|s;\theta) + \pi(a|s;\theta)\nabla_\theta Q_\theta(s,a)\right)\right] \\
&= \mathbb{E}_{s,a \sim \rho_\theta}\left[\gamma(s)\left(Q_\theta(s,a)\nabla_\theta \log \pi(a|s;\theta) + \nabla_\theta Q_\theta(s,a)\right)\right] \\
&= \mathbb{E}_{s,a \sim \rho_\theta}\left[\gamma(s)Q_\theta(s,a)\nabla_\theta \log \pi(a|s;\theta)\right] + \mathbb{E}_{s,a \sim \rho_\theta}\left[\gamma(s)\nabla_\theta Q_\theta(s,a)\right].
\end{aligned}
\tag{25}
$$

Moving the second term in the right-hand side of (25) to the left, yields

$$
\mathbb{E}_{s,a \sim \rho_\theta}\left[\left(1 - \gamma(s)\right)\nabla_\theta Q_\theta(s,a)\right] = \mathbb{E}_{s,a \sim \rho_\theta}\left[\gamma(s)\, Q_\theta(s,a)\nabla_\theta \log \pi(a|s;\theta)\right]
\tag{26}
$$

Now consider the specific $\gamma$ function with $\gamma(s) = 1$ for $s \notin \mathcal{S}_\perp$, and $\gamma(s) = 0$ for $s \in \mathcal{S}_\perp$. Substituting such episodic $\gamma$ function into (26), yields

$$
\sum_{s \in \mathcal{S}_\perp} \rho_\theta(s) \mathbb{E}_{a \sim \pi(s;\theta)}\left[\nabla_\theta Q_\theta(s,a)\right] = \sum_{s \notin \mathcal{S}_\perp} \rho_\theta(s) \mathbb{E}_{a \sim \pi(s;\theta)}\left[Q_\theta(s,a)\nabla_\theta \log \pi(s,a;\theta)\right].
\tag{27}
$$

The left-hand side of (27) is an average of $\nabla_\theta Q_\theta(s,a)$ over terminal states $s \in \mathcal{S}_\perp$ and over the policy-induced actions $a \sim \pi(s;\theta)$, but note that $\nabla_\theta Q_\theta(s_\perp, a) = \nabla_\theta J_{epi}(\theta)$ for any such $s \in \mathcal{S}_\perp$ and any $a \in \mathcal{A}$, which

means at the left-hand side we are averaging a constant term that can be moved out of the summations (of both $s$ and $a$). Further denoting $\rho_\theta(\mathcal{S}_\perp) \doteq \sum_{s \in \mathcal{S}_\perp} \rho_\theta(s)$, yields

$$
\begin{aligned}
\nabla_\theta J_{epi} &= \frac{1}{\rho_\theta(\mathcal{S}_\perp)} \cdot \sum_{s \notin \mathcal{S}_\perp} \rho_\theta(s) \underset{a \sim \pi(s;\theta)}{\mathbb{E}} \left[ Q_\theta(s,a) \nabla_\theta \log \pi(s,a;\theta) \right] \\
&= \frac{\rho_\theta(\mathcal{S} \setminus \mathcal{S}_\perp)}{\rho_\theta(\mathcal{S}_\perp)} \cdot \sum_{s \notin \mathcal{S}_\perp} \frac{\rho_\theta(s)}{\rho_\theta(\mathcal{S} \setminus \mathcal{S}_\perp)} \underset{a \sim \pi(s;\theta)}{\mathbb{E}} \left[ Q_\theta(s,a) \nabla_\theta \log \pi(s,a;\theta) \right] \qquad (28) \\
&= \frac{1 - \rho_\theta(\mathcal{S}_\perp)}{\rho_\theta(\mathcal{S}_\perp)} \cdot \underset{s,a \sim \rho_\theta}{\mathbb{E}} \left[ Q_\theta(s,a) \, \nabla_\theta \log \pi(s,a;\theta) \, \middle| \, s \notin \mathcal{S}_\perp \right]
\end{aligned}
$$

Finally, due to Proposition 4, $\frac{1 - \rho_\theta(\mathcal{S}_\perp)}{\rho_\theta(\mathcal{S}_\perp)} = \mathbb{E}_\theta[T] - 1$.                    $\square$

The proofs of Theorem 4 and 5 demonstrates a pattern for steady state analysis in episodic tasks: first examine properties of *any* RL task under its steady state, then substitute in a specific $\gamma$ function to derive corollaries dedicated to episodic tasks as special cases. The same approach could potentially be used in more theoretical and algorithmic problems of episodic RL.

# D   On the definition of on-policy distribution in episodic tasks

In page 199 of Sutton and Barto [23], the on-policy distribution of episodic tasks has been defined as $\mu_\pi^1$, the normalized undiscounted expected visiting frequency of a state in an episode. We believe the stationary distribution $\rho_\pi$ identified in this paper helps with better shaping this conception in at least three aspects:

First, defining on-policy distribution directly as the stationary distribution $\rho_\pi$ helps unify the notion of on-policy distribution across continual and episodic tasks without changing its mathematical identity (as $\rho_\pi = \mu_\pi^1$).

Second, it seems that the conception of on-policy distribution is *intended* to capture "*the number of time steps sent, on average, in state $s$ in a single episode*" (Sutton and Barto [23], page 199). However, note that there are two possible formal semantics: $\mathbb{E}_\varsigma[n_s]/\mathbb{E}_\varsigma[T]$ and $\mathbb{E}_\varsigma[n_s/T]$ – both capture the intuition of "time spent on $s$ in an episode on average", and it is not immediately clear why we should favor normalizing all episodes uniformly by the average episode length over normalizing in a per-episode manner. In fact, the ergodic theorem (i.e. Proposition 3) has "favored" the latter semantic, by connecting the per-episode-normalized reward $\mathbb{E}[\sum_{t=1}^T R(s_t)/T]$ to the steady-state reward $\mathbb{E}_{s \sim \rho_\pi}[R(s)]$. In comparison, Theorem 4 "favors" the former semantic, establishing equality between the uniformly-normalized distribution $\mu_\pi^1$ and the steady-state distribution $\rho_\pi$, which justifies formal semantic of ambiguous intuition in a more principled way.

Lastly, in Sutton and Barto [23] (page 199, Eq. 9.2), the "existence" of the on-policy distribution is defended by writing $\mu_\pi^1$ into the (normalized) solution of a non-homogeneous system of linear equations. But not every non-homogeneous linear system has a unique solution. Also, the linear-system argument becomes more subtle when generalizing from finite state spaces (which is assumed in [23]) to infinite state spaces. Our treatment to the stationary distribution $\rho_\pi$ thus consolidates the concept of on-policy distribution by providing an alternative theoretical basis (based on the markov chain theory) for this term.

# E   On what traditional policy-gradient estimators actually compute

As common practice, traditional policy gradient algorithms use the discounted value function but compute the gradient based on undiscounted data distribution, which neither follows the classic (discounted) policy gradient theorem nor does it follow the undiscounted steady-state policy gradient theorem. This gap between theory and practice is well known in the community, and [27] has examined what such "mixed" policy gradient estimation would obtain in the continual settings. But similar analysis in the episodic case was not done exactly because of the lack of steady state for the latter. As Thomas [27] noted, "*[the stationary-rewarded objective] $\bar{J}$ is not interesting for episodic MDPs since, for all policies, [the stationary distribution] $\bar{d}^\theta(s)$ is non-zero for only the post-terminal absorbing state. So, henceforth, our discussion is limited to the non-episodic setting*".

However, the existence of unique stationary distribution now enables such analysis even for episodic tasks. Specifically, substituting $\gamma(s) = \gamma_c < 1$ into (24), which is copied below for convenience

$$
\underset{s,a \sim \rho_\theta}{\mathbb{E}} \left[ \left( 1 - \gamma(s) \right) \nabla_\theta Q_\theta^{\gamma_c}(s,a) \right] = \underset{s,a \sim \rho_\theta}{\mathbb{E}} \left[ \gamma(s) Q_\theta^{\gamma_c}(s,a) \nabla_\theta \log \pi(a|s;\theta) \right],
$$

yields,

$$
\underset{s,a \sim \rho_\theta}{\mathbb{E}} \left[ Q_\theta^{\gamma_c}(s,a) \nabla_\theta \log \pi(a|s;\theta) \right] = \frac{1 - \gamma_c}{\gamma_c} \underset{s,a \sim \rho_\theta}{\mathbb{E}} \left[ \nabla_\theta Q_\theta^{\gamma_c}(s,a) \right] \qquad (29)
$$

The left-hand side is exactly what the classic policy gradient algorithms compute in practice, and the right-hand side is proportional to the gradient of the steady-state performance *if* the policy change does not affect the stationary distribution. This observation is in analogy to what Thomas [27] concluded in the continual setting.

# F   Experiment details

This section complements Section 6 to report more detailed experimentation settings and results. In all experiments, the behavior policy is a Gaussian distribution with diagonal covariance, whose mean is represented by a neural network with two fully-connected hidden layers of 64 units[17]. All model parameters are randomly initialized. The RoboSchoolBullet benchmark [1] consists of challenging locomotion control tasks that have been used as test fields for state-of-the-art RL algorithms [7].

## F.1   Details of the gradient checking experiments

In the gradient checking experiment reported in Section 6.1, we run a standard stochastic gradient ascent procedure with the gradient estimated by $F_{SSPG}$. In each policy-update iteration, we run $K$ independent rollouts, each lasts for ten episodes. As our focus is to examine the quality of gradient estimation, we set the batch size $K$ to be one million, so as to have accurate estimate of the ground-truth gradient. We then compute the AEL term $\widehat{\mathbb{E}_\theta[T]} - 1$ by averaging over the $10 \cdot K$ episodes, which is very close to the true AEL value (minus 1) thanks to the very large sample size. The next step is to collect the $(s_t, a_t)$ pair at the single step of $t = 3 \cdot AEL$ in each of the $K$ rollouts, creating an i.i.d. sample $\mathcal{D}$ of size $K$. The $\widehat{Q_\theta}(s, a)$ term for $(s, a) \in \mathcal{D}$ is estimated using the corresponding episode return (from $s$).

Two gradient estimators are implemented: one follows exactly $F_{SSPG}$ with the AEL term, the other omits the AEL term from $F_{SSPG}$. We applied constant learning rate $\alpha = 5 \times 10^{-4}$ to the estimator *omitting* the AEL term, which simulated what traditional policy gradient estimators have been doing. The resulted policy update from this baseline method is thus $\Delta\theta = \alpha \cdot \bar{F}_{SSPG}/(\mathbb{E}[T] - 1)$. We call such a "standard practice" of policy-gradient update, the *constant learning rate* method. It is clear that the "constant learning rate" method is essentially applying a drifting learning rate $\alpha/(\mathbb{E}[T] - 1)$ to the truly unbiased gradient estimator $\bar{F}_{SSPG}$.

On the other hand, we applied the same constant learning rate $\alpha$ also to the estimator *with* the AEL term, computing policy update as $\Delta\theta = \alpha \cdot \bar{F}_{SSPG}$. This is equivalent to applying an AEL-adaptive learning rate $\alpha \cdot (\mathbb{E}[T] - 1)$ to the traditional policy gradient estimator (that omits the AEL term), and is thus called the *adaptive learning rate* method.

We measure the quality of an estimated policy gradient $\widehat{\nabla_\theta J}$ by examining the quality of the projected performance change $\widehat{\Delta J}$ as entailed by the estimated gradient. Specifically, let $\Delta\theta = \widehat{\nabla_\theta J} \cdot \alpha$ be the corresponding policy update of the estimated gradient, the projected performance change from such a policy update is calculated as $\widehat{\Delta J} = ||\widehat{\nabla_\theta J}||_2 \cdot ||\Delta\theta||_2 = ||\widehat{\nabla_\theta J}||_2^2 \cdot \alpha$. It is known that for the $\widehat{\Delta J}$ thus computed, we have $\widehat{\Delta J} = \Delta J$ if $\widehat{\nabla_\theta J} = \nabla_\theta J$, and that the more biased the estimated policy gradient $\widehat{\nabla_\theta J}$ is (to the true gradient $\nabla_\theta J$), the more biased the projected performance change $\widehat{\Delta J}$ will be (to the true performance change $\Delta J$). Based on this principle, we computed, for each policy-update iteration, the projected performance changes from both the "constant learning rate" method and the "adaptive learning rate" method, and compared them with the true performance change $\Delta J = J_{new} - J_{old}$ whose %90-confidence interval is calculated from the statistics of episode-wise returns in the $10 \cdot K$ independent rollouts, for both the old and new policies.

We conducted the above experimentation procedure to the HopperBulletEnv-v0 environment in RoboSchool-Bullet, and Figure 1 revealed how quickly the drifting AEL term can hurt the quality of gradient estimation in this environment. The shaded area illustrates the $90\%$ confidence intervals of the true performance changes after each policy update. The red dotted curve is the projected performance change from the "constant learning rate" method which treats the AEL term as "a proportionality constant that can be absorbed in the learning rate". We see that this traditional policy gradient method quickly leads to bias after only tens of iterations, as Figure 1(a) shows. The bias becomes quite significant after 100 updates, as Figure 1(b) shows. On the other hand, the orange curves are the projected performance change by the "adaptive learning rate" method, which follows the unbiased estimator as given by Theorem 5, and leads to much less bias as Figure 1(a) and Figure 1(b) show.

## F.2   Details of the perturbation experiments

Figure 2a shows how the mean value of the marginal distribution for each state dimension evolves over time in the (multi-episode) learning process of the *raw* Hopper environment under random policy, without perturbation. Each mean-value point in the figure is calculated by averaging over 100,000 rollouts, which serves as an index, or an indicator to the marginal distribution for the corresponding state dimension. Time is normalized to multiples of Average Episode Length (or AEL), and we see that the marginal distributions of all state dimensions have

(a) Policy iteration $0 - 30$        (b) Policy iteration $100 - 130$

Figure 1: Projected vs actual performance changes in HopperBulletEnv-v0 as quality checking for policy gradient estimators with the AEL term (corresponding to "adaptive learning rate") and without the AEL term (corresponding to "constant learning rate").

converged at $t = 2 \cdot AEL$. We found that similar convergence rates apply to policies at different stage of the RL training in Hopper. The result shows that perturbation may not be necessary for the Hopper environment.

On the other hand, Figure 2c and 2e show that the marginal distributions may converge much slower in some other environments. In the Humanoid environment, the marginal distribution of the state dimensions takes more than $20 \cdot AEL$ steps to converge, while in the HalfCheetah environment, the marginal distribution appears to not converge at all. Both environments indeed have strong periodicity in episode length. Especially, the episode length of Halfcheetah is actually fixed at exactly 1000 under the random policy (and under any other policy as well). These observations indicate that perturbation is indeed needed in general, if we want to approximate the stationary distributions with marginal distributions of a single (or a few) step.

Moreover, Figure 2d and 2f illustrate that the rollouts in Humanoid and HalfCheetah quickly converge to their respective steady states after applying the recursive perturbation trick with $\epsilon = 1 - 1/\mathbb{E}[T]$. Comparing with Figure 2c and 2e, respectively, we clearly see the effectiveness of the recursive perturbation on these two popular RL environments. In both cases the convergence occurs before the step of $3\,\mathbb{E}[T]$.

Figure 2b shows that the "3-AEL convergence" observation generalizes to even adversarially synthesized environments. These environments have fixed episode lengths $n$ and the state $s_t$ regularly goes over from 0 to $n - 1$ in an episode. Without perturbation, the marginal distribution $\rho^{(t)}$ of such environment would concentrate entirely on the single state $(t \mod n)$. We then applied the recursive perturbation to state-sweeping environment with $n = 20, 100, 500, 2000$. In all cases $\epsilon = 1 - 1/n$, and we run a large number of independent (and perturbed) rollouts for marginal distribution estimation (to observe the ground-truth distribution, we have to run 30 million rollouts when $N = 2000$). In each of the rollouts we collected the states at $t = 3n$ and $t = 3n-1$ as two sample points, and we observed the empirical state distribution of all the samples thus obtained, for each environment (i.e. for each $n$). As we can see from the figure, even for the completely periodic environment with fixed episode length $n = 2000$, the marginal distribution still converges well to the stationary distribution (which is the uniform distribution over $\{0 \ldots n - 1\}$) in only $3n$ steps, after applying the recursive perturbation.

Note that the perturbation with self-loop probability $\epsilon = 1 - 1/n$ causes the rollout to stay in the null state for $n - 1$ steps per episode on average, which in turn causes half of the samples obtained at a fixed rollout time to be the null state (if the marginal distribution at that time has already converged). We sampled two consecutive time steps around $3n$ to compensate this loss of data, so that the two-step sampling provides roughly the same amount of "useful samples" as the original batch size (in one-step sampling without perturbation). Again, as Figure 2b illustrates, the empirical distribution from such two-step sampling approximates the desired uniform distribution pretty well around $t^* = 3n$. In other words, the practical cost for the half amount of samples wasted in the null state is to just sample one more step in the rollout.

(a) HopperBulletEnv-v0 (raw)

(b) State-Sweeping (perturbed)

(c) HumanoidBulletEnv-v0 (raw)

(d) HumanoidBulletEnv-v0 (perturbed)

(e) HalfCheetahBulletEnv-v0 (raw)

(f) HalfCheetahBulletEnv-v0 (perturbed)

Figure 2: The "3-AEL convergence" phenomenon