[Reviews · NeurIPS 2020]

Review 1

Summary and Contributions: This paper takes a theoretical looks at the link between episodic RL and RL in continuing environments. It is demonstrated that with a slight perturbations to the conventions of the MDP, it can be proven that episodic RL problems are ergodic. The implications for the policy gradient over the steady-state distribution are analyzed. Additionally, in the appendix the mixing time properties for environments from the literature and the gradient estimate quality is empirically analyzed.

Strengths: I found this paper to be an enjoyable read and quite interesting. The conclusion that episodic RL tasks are nearly always ergodic in my opinion is very insightful, opening up the doors to new viewpoints on results in this setting from a steady-state perspective. The authors provide a good example in the appendix where they analyze the mixing time properties of some environments from the literature. The paper is clearly written and does a good job taking the reader step by step through each result. I thought the point made at the end coming out of this work about how actually continuing environments are less likely to be ergodic than episodic environments was eye opening.

Weaknesses: I personally find this topic very interesting, but admittedly the topic of ergodicity is quite theoretical and the paper will have a limited audience as a result. The insights do not directly lead to an approach that can be empirically validated. As such, the paper could potentially be improved by spending more time discussing the potential future impact and implications of the derived results so that it is clear to a wider audience.

Correctness: The empirical analysis is only contained in the appendix. However, I found the results there quite relevant and interesting. It seems that results have been averaged over many runs to promote reproducibility. I have not thoroughly gone through the proofs of all 5 theorems, but the proof ideas did make sense to me.

Clarity: I thought that this paper was very well written and clear as a whole. It takes on a pretty dense subject, but really goes through the analysis step by step in a way that I found quite straightforward to follow.

Relation to Prior Work: Yes. It seems that while prior work has considered that episodic RL is a special case of continuing RL, it is a novel result to show that episodic problems are nearly always ergodic thus yielding a unique steady-state distribution.

Reproducibility: Yes

Additional Feedback: After reading the other reviews and author feedback, I still feel that my initial score is appropriate and that this paper should make a nice contribution to the conference.


Review 2

Summary and Contributions: The main contribution of this paper is that it proves a unique stationary distribution exists in every episodic learning process. Based on that, a formal relation between the stationary reward objective and the episode return objective is presented, and a steady-state policy gradient theorem for episodic learning processes is provided.

Strengths: This paper makes an interesting theoretical contribution to the understanding of reinforcement learning task specification. The paper is well written and should be easy to follow for a broad audience.

Weaknesses: In general, I think this is a good theoretical paper (but my confidence is low as I am not very familiar with this field). I only have some minor comments about its significance. While the contributions are primarily from a theoretical side, are the results also of practical interest? This paper claims that with the existence of steady-state distribution, one can first examine the properties of episodic learning tasks under the steady state, and then derive dedicated corollaries. However, I don't see clearly how this may help except the proposed Theorem 5. The significance could be further improved, if the paper addressed how the theorems shed light on the design of new learning algorithms, point out what is missing in existing algorithms, etc. ----------------------Updates after the rebuttal---------------------------------------------- The authors' rebuttal show that their theoretical results may also have some practical impacts, which addresses my concern to some degree. I suggest that the author make some room for such kind of discussion in the revised version. This will make this work interesting and relevant to a broader readership.

Correctness: I didn't check the proofs, but the theoretical claims look ok.

Clarity: Yes. The paper is well written and easy to follow.

Relation to Prior Work: Yes.

Reproducibility: Yes

Additional Feedback:


Review 3

Summary and Contributions: The authors consider a transformation from an episodic problem (finite horizon) to a continuing problem (infinite horizon) with extending the ideas of unifying task specification [25] and the product-form value functions [20] and define the transformed continuing problem as the episodic learning process. Since the episodic learning process is proper infinite horizon MDP, they almost directly derive the results known in infinite horizon MDP. Although the presented results might be known implicitly in the RL experts, these have not been discussed in detail so far. I think this type of contribution about unification is important because I think this type of contribution about task unification or simplification is important because there are a lot of RL task variations. Due to this work, we could use RL algorithms properly. This work helps us to use/select the RL algorithms more comfortably and properly.

Strengths: I found the following strengths in the paper. * Gives a useful transformation from an episodic process (finite horizon) to a continuing process (infinite horizon). * Givens useful properties of the transformed process, which are corresponding to them known in infinite horizon RL. * Gives a perturbed MDP model for ensuring aperiodicity.

Weaknesses: The following are the weaknesses that I have noticed. * The presented results seem to be straightforward and would be known implicitly in the RL experts. (But these have not been discussed in detail so far) * It is still unclear how the results can contribute to the development of the algorithm.

Correctness: I did not find out serious incorrect part.

Clarity: The paper is well written so that I could follow most of it easily.

Relation to Prior Work: To the best of my knowledge, the authors make a proper reference to the prior work and clearly discuss how their work extends the prior work.

Reproducibility: Yes

Additional Feedback: I have the following minor comments and questions. * Introduction seems to be biased to the episodic RL, which might be misleading especially for non RL-experts. I believe that both episodic learning and continual learning are equally important. The appropriate learning type depends on the task. * Line 111: E_{\zeta\sim\rho_0,…} should be E_{s_0\sim\rho_0,…}? * It is helpful to explicitly define the episodic gamma-discounting in Line 294. * The aperiodicity is really needed for the policy gradient theorem of episodic learning process, etc.? As the authors described in Line 275-276, the perturbation for aperiodicity could be omitted in reality.


Review 4

Summary and Contributions: The paper investigates the learning dynamics of episodic reinforcement learning (RL). The authors define a class of Markov decision processes (MDPs) called episodic learning processes which models the usual episodic formulation of RL. They then proceed to show that this class of MDPs is ergodic, which has several positive implications for learning. First, since now every policy has a stationary distribution, this provides a more clear, unified, definition of the term “on-policy distribution” (and also connects it with the definition used in continual RL). Second, the authors show that the usual practice of estimating the policy gradient using a single sample transition is sound under mild assumptions. Third, their analysis also sheds light on a well known discrepancy between theory and practice in policy gradient algorithms: namely, the fact that these methods use the discounted value function but the gradient is computed based on the undiscounted distribution.

Strengths: Modelling and analyzing the learning dynamics of episodic RL is a fundamental question that is of much interest to the NeurIPS community.

Weaknesses: As discussed below, the significance and implications of the theoretical findings is obfuscated by the excessive use of technical jargon and a slightly convoluted presentation.

Correctness: The theoretical results seem to be correct; although I have not checked the proofs in the supplementary material, the description of the “proof ideas” in the main paper look sensible (and, more generally, the claims make intuitive sense). The only empirical evaluation is in the supplementary material, and I did not check it carefully.

Clarity: In my opinion the main weakness of the paper is its presentation. First, there is a lack of clear, direct, explanations of what the paper is trying to accomplish. Several crucial points are either only implied or mentioned in passing without the proper emphasis. This is true for the positioning of the paper itself. The analysis seems to be mostly concerned with policy gradient methods, but this is never explicitly stated. Value-based methods are neither excluded nor directly discussed. There is also a lack of clarity regarding the implications of the theoretical results. Take Theorem 4, for example. In Section 4 it is shown that an episodic learning process is always ergodic. This means that they automatically inherit the property in Propositions 3, namely, that the average return per episode approaches the expected policy return in the limit. But then, in Theorem 4, it is shown that in episodic learning processes the mean return divided by the mean episode length equals the expected policy return. This seems like an interesting result, but the text never explicitly discusses its implications or why it is important. A similar observation applies to Theorem 5, though to a lesser extent. In both cases the text focuses on pinpointing differences with previous results or algorithms, leaving to the reader the burden of figuring out the consequences of such differences. I believe the observation above is the reflex of the style of the narrative more broadly. I have the impression that most of the information is there, but sometimes conclusions are hidden behind technical observations whose implications are assumed to be obvious to the reader. In my opinion this paper would greatly benefit from some rewriting to make sure conclusions are clearly spelled out.

Relation to Prior Work: Yes. The most relevant reference seems to be White’s “Unifying task specification in reinforcement learning”, which is discussed in the paragraph starting on line 352.

Reproducibility: Yes

Additional Feedback: POST-REBUTTAL COMMENTS: ------------------ Thank you for the clarifying rebuttal! After reading it and the other reviews, as well as revisiting the paper and looking at the supplement more carefully, I decided to increase my score to 6. The idea of "connecting" the terminal states of an episodic MDP with the initial distribution is not new (see, for example, White, 2017). But, as the authors point out, the paper does take this idea one step further, analyzing the properties of the induced MDP. In doing so they arrive at some interesting insights, which provide theoretical support for some algorithms used in practice (Section E of the supplementary material) and also suggest some new tricks (Section F). I suspect the former will have more of an impact as a contribution than the latter. As I said in my review, my main concern regarding this paper is its presentation. As is, it comes across as a compressed version of a longer manuscript, and as a reader I could see that some material had to be cut. Looking at the supplement and the rebuttal, I think the authors have enough material to put together a good paper, but that will, in my opinion, require restructuring things a bit. For example, the authors spend quite some time explaining the basics of Markov chains and MDPs in Section 2, including different formulations of the latter, at the expense of explaining some of their own insights in more detail --which are deferred to the appendix. I urge the authors to take this comment seriously and to put some effort in making the next version of the paper more clear, otherwise an interesting contribution might end up obfuscated by the presentation.

[Author Response · NeurIPS 2020]

We thank all reviewers for the many helpful comments to our paper. A common suggestion from most reviewers is to
further elucidate how our theory helps develop better RL algorithms. We highlight two such algorithmic ideas below.

**Recursive perturbation**. Figure 1(e) in the appendix showed that in the Humanoid environment, the marginal distri-
butions of $s_t$ may take as many as $20 \cdot \mathbb{E}[T]$ steps to converge to the steady state (where $\mathbb{E}[T]$ is the average episode
length), while Figure 1(f) showed that the marginal distributions of HalfCheetah do not converge at all (because
$\mathbb{E}[T] \equiv 1000$ in HalfCheetah, a situation where the marginal distributions will be strongly phased, as discussed in
Section 4). These observations illustrate the *necessity* to apply the $\epsilon$-perturbation trick in real-world RL practice.

In Section F.2 of the appendix we introduced an augmented variant of $\epsilon$-perturbation. The idea is to recursively
apply the one-shot perturbation to already perturbed models, which in the limit is equivalent to a perturbation with
self-looping probability $\epsilon$. We empirically found that recursive perturbation with $\epsilon = 1 - 1/\mathbb{E}[T]$ appears to force
convergence of the marginal distribution at $t = 3\mathbb{E}[T]$ in *various* environments. Figure 2(b) in the appendix already
showed such a result on a synthetic but challenging environment. Figure (A) and (B) below further demonstrate the
same observation in Humanoid and HalfCheetah. Comparing Figure (A) with Figure 1(e), and Figure (B) with Figure
1(f), we can see clearly the effectiveness of the recursive perturbation on these two popular RL environments.

**Episode-length-adaptive policy gradient**. Our steady-state policy gradient theorem (i.e. Theorem 5) showed a pro-
portionality factor of $\mathbb{E}_\pi[T] - 1$ between the true policy gradient $\nabla J_{epi}$ and the classic policy gradient estimator
$F(\theta) = \mathbb{E}_{s,a \sim \pi}[Q_\pi(s,a)\nabla \log \pi(s,a;\theta)|s \notin \mathcal{S}_\perp]$, and we have argued in Section 5 (line 320-326) that this pro-
portionality factor can change dramatically in practical RL training, and that while it will not change the gradient
direction, using $F(\theta)$ to estimate the policy gradient is equivalent to applying a learning rate $\beta = \frac{\alpha}{\mathbb{E}_\pi[T]-1}$ to the truly
unbiased estimator, where $\beta$ is dynamically drifting along with the changes of episode length during the training.

Figure (C) and (D) above revealed how quickly the drifting episode length can hurt the quality of gradient estimations
in the Hopper Environment. As with Figure 3 in appendix, the shaded area gives $90\%$ confidence intervals of the true
changes of policy performance in iterations with $\alpha = 5 \times 10^{-4}$. The red dotted curve is the predicted performance
change by assuming the proportionality factor as a constant absorbed in the learning rate, which quickly leads to bias
after only tens of iterations, as Figure C shows. The bias becomes quite significant after 100 iterations, as Figure
(D) shows. On the other hand, the orange curves are the estimated performance change by following the unbiased
estimator given by Theorem 5, which leads to much less bias (see Section F.3 for more details about this experiment).

Meanwhile, we try to give in this response a better **summary of the theoretical implications** of our paper:

1. **(a)** We identified two formal properties (Definition 1) in the learning environments of finite-horizon decision tasks.
**(b)** We proved that these properties imply existence of nondegenerate stationary distribution *in finite-horizon tasks*.
2. **(a)** We proposed a perturbation trick (one-shot version in Definition 2, recursive version in Section F.2) and proved
that the perturbed model (of learning environment of finite-horizon task) is ergodic. **(b)** The *guaranteed* (instead
of assumed) ergodicity implies that few-step sampling is sound in *the perturbed model of* all finite-horizon tasks.
3. **(a)** We analyzed the Bellman equation under the steady state, which connects, for *any* function, its mean values over
the state space and over the episode space. **(b)** As two special cases of 3(a), we showed $J_{epi}(\pi) = J_{avg}(\pi) \cdot \mathbb{E}_\pi[T]$
and $\rho_\pi = \mu^1_{epi}$, which in turn help unify RL formulations between continual and episodic tasks (Table 1).
4. **(a)** We analyzed the differentiated Bellman equation under the steady state, which leads to $\nabla J_{epi} = (\mathbb{E}_\theta[T] - 1) \cdot$
$\mathbb{E}_{s,a \sim \rho_\theta}[\dots]$. **(b)** 4(a) implies two sources of bias in existing policy gradient algorithms as discussed above, i.e. the
fluctuating marginal distribution in tasks like HalfCheetah and the drifting proportionality factor.

*To Reviewer 3*: We fully agree that continual and episodic RL are equally important, yet please note that the eight
results of this particular paper (see the list above) apply only to episodic RL with finite horizon. Although we did use
infinite-horizon MDP as an *approach*, the *problems* we solved seem to have limited overlap with infinite-horizon RL.

*To Reviewer 4*: We hope the summary above better explained what the paper accomplished. The positioning of the
paper is actually general, with 6 out of the 8 result items applicable to both policy-based and value-based methods
(e.g. parallel few-step sampling was used in both methods [14]). Please see Section B for more discussion on this.

[Meta-Review · NeurIPS 2020]

This paper provides a new perspective in thinking about episodic RL, and should be of interest to anyone working with MDPs in reinforcement learning. Three reviewers (R1, R2, R3) commented that it was well-written and clear, although R4 disagreed. All reviewers commented on the interesting contributions (proving that MDPs within episodic RL can be proven to be ergodic). R1, R2, and R3 had concerns that it was a mostly theoretical paper, and wondered how to practically apply these insights. However, the rebuttal goes some way to address these points, and R4 was convinced to raise their recommendation to weak accept. I think these kinds of more theoretical, analytical papers are pivotal toward increasing understanding of RL models and how they learn, and all reviewers agree it’s a very well-presented and motivated paper along these lines. I therefore recommend accept.